psychology/health and disease and epidemiology

pandemic, perceived risk, lockdown, protection motivation theory, distress

**Author for correspondence:**
Gerit Pfuhl
e-mail: gerit.pfuhl@uit.no

†Shared first-authorship; these authors contributed equally to this study.

# Perceived efficacy of COVID-19 restrictions, reactions and their impact on mental health during the early phase of the outbreak in six countries

Martin Jensen Mækelæ[1,†], Niv Reggev[2,†], Natalia Dutra[3], Ricardo M. Tamayo[4], Reinaldo A. Silva-Sobrinho[5], Kristoffer Klevjer[1] and Gerit Pfuhl[1]

[1]Department of Psychology, UiT The Arctic University of Norway, Tromsø, Norway
[2]Department of Psychology and Zlotowski Center for Neuroscience, Ben Gurion University of the Negev, Israel
[3]Department of Physiology and Behavior, Universidade Federal do Rio Grande do Norte, Brazil
[4]Departamento de Psicología, Universidad Nacional de Colombia, Colombia
[5]Laboratory of Epidemiology and Operational Research in Health, Western Paraná State University – Unioeste, Foz Iguaçu-PR, Brazil

MJM, 0000-0002-6791-1218; NR, 0000-0002-5734-7457; ND, 0000-0002-0766-0795; RMT, 0000-0002-8678-0145; RAS-S, 0000-0003-0421-4447; KK, 0000-0003-4298-4418; GP, 0000-0002-3271-6447

The COVID-19 pandemic forced millions of people to drastically change their social life habits as governments employed harsh restrictions to reduce the spread of the virus. Although beneficial to physical health, the perception of physical distancing and related restrictions could impact mental health. In a pre-registered online survey, we assessed how effective a range of restrictions were perceived, how severely they affected daily life, general distress and paranoia during the early phase of the outbreak in Brazil, Colombia, Germany, Israel, Norway and USA. Most of our over 2000 respondents rated the restrictions as effective. School closings were perceived as having the strongest effect on daily life. Participants who believed their country reacted too mildly perceived the risk of contracting SARS-CoV-2 to be higher, were more worried and expressed reduced beliefs in the ability to control the outbreak. Relatedly, dissatisfaction with governmental reactions corresponded with increased distress

levels. Together, we found that satisfaction with one's governmental reactions and fear appraisal play an important role in assessing the efficacy of restrictions during the pandemic and their related psychological outcomes. These findings inform policy-makers on the psychological factors that strengthen resilience and foster the well-being of citizens in times of global crisis.

# 1. Introduction

On the 11 March 2020, COVID-19 was officially classified as a global pandemic [1]. The COVID-19 outbreak, starting in late 2019 and lasting throughout the time the present manuscript was written, poses severe risks for the physical and mental health of people around the world. As of 18 June 2020, the total number of deaths was 445 535, of a total of 8 242 999 confirmed cases [1]. In certain countries, the pandemic had also led to a partial and temporary collapse of the healthcare system due to the high rate of patients requiring medical treatment [1]. These prospective risks led governments around the world to enact various restrictions in an attempt to contain the pandemic, ranging from public guidelines of social distancing (e.g. as in Sweden; Public Health Agency of Sweden) to complete lockdowns (e.g. for the district of Wuhan, China; [2]).

One key factor for adherence with public health advice, e.g. social distancing, is the perception of their efficacy (beyond feeling threatened) [3]. In this context, knowing that those restrictions are effective in protecting ourselves and that we are able to follow that advice will boost our self-efficacy. Accordingly, in the current investigation, we aimed to assess the perceived efficacy of these restrictions and their perceived impact on everyday life.

Furthermore, the impact of a pandemic and the measures to restrict it on mental health tend to outlast their physical effects, and has important psycho-social and economic consequences [4–7]. It is now undeniable that the restrictions against COVID-19 drastically changed the lifestyle of most members of present-day societies [1], potentially contributing to the pandemic-related mental health burden. We therefore assessed how the perceived risk of COVID-19 and the various restrictions affected psychological well-being of participants at an early phase of the outbreak. Finally, we planned the current measures with a longitudinal outlook to allow us to collect additional data in the near future.

## 1.1. The effects of sudden changes in external circumstances on mental health

Engaging in an abrupt change in everyday life inevitably leads individuals to experience a heightened sense of personal and societal uncertainty [8,9]. Uncertainty, in turn, can lead to numerous detrimental impacts on one's well-being [10,11]; for potential positive consequences such as increased prosociality (e.g. [12,13]). Such a sudden change in everyday life is particularly true in the context of the early stages of the COVID-19 outbreak, as the WHO called all countries to apply strict restrictions (e.g. global surveillance, quarantine, isolation facilities and control practices) in an attempt to reduce human-to-human transmission [1]. The potential impacts and accompanying uncertainty of these measures are unprecedented in recent history [14]. Interestingly, as individuals are generally poor predictors of their future effect ([15,16] but see [17]), people who have yet to experience COVID-19-related restrictions should experience the impact of these measures as less severe than people already experiencing them.

The restrictions taken to contain the outbreak and the uncertainty associated with their implementation can have multiple detrimental effects on psychological well-being. For example, the (publicly) acknowledged uncertainty about the outbreak can reduce the feeling of being able to control the outbreak, can lead people to misjudge the probability of contagion [18], and can increase psychological distress [19]. These, in turn, can lead to anxiety and paranoia [20–22] and lower intentions to engage in protective health behaviour [23,24]. Moreover, these restrictions have drastically altered the very fabric of human existence, banning most forms of social interactions and requiring individuals to rapidly adapt to completely different daily routines. Indeed, many countries have introduced partial or full lockdowns, including quarantines and restrictions of various degrees to many parts of their population. However, social isolation and loneliness carry a negative impact on both mental and physical health [25,26], and quarantine may cause stress symptoms, as well as anger and confusion [27]. Prolonged stress, in turn, can lead to aberrant regulation of the immune system and higher susceptibility to virus infections [28]. These negative impacts can also extend to damage to the fabric of society, as paranoia and related beliefs in conspiracy theories can motivate criminal activity [29]. Together, loneliness, loss of control, distress, perceived risk and fear of contagion can negatively impact physical and mental health

as well as cause social harm. Thus, the current investigation explored the contributions of these factors to psychological well-being.

## 1.2. Perceived efficacy of restrictions and reactions

One important resilience factor that can potentially attenuate the detrimental psychological effects of governmental restrictions is their perceived efficacy and the individual coping response efficacy, e.g. protection motivation theory [23,24,30]. Perceptions of governmental actions as effective promote life satisfaction and support protective behaviour [3,24]. Similarly, positive appraisal of public health policies fosters a positive social climate [31]. Contemporary models of large-scale behavioural interventions designed to prevent diseases prescribe a key role to fear and perceived risk in the perceived efficacy of these interventions [32]. For instance, graphic health warnings on cigarette packages, endorsed by the WHO as a useful tool to prevent smoking initiation and promote quitting, work via triggering fear of lung cancer [33]. Similarly, for the goals of the present study, we considered perceived risk and fear as potential contributors to individuals' perceptions about the efficacy of restrictions. Specifically, we predicted that the higher an individual would estimate the probability of contagion and lethality for COVID-19, the more they will judge severe restrictions to personal freedom as effective.

The perceived efficacy of restrictions should theoretically also relate to participants' epistemological state. Factual and rational information about diseases fosters consistent and healthy behaviours as well as intentions to endorse healthy habits [31,34]. For example, dietary recommendations are usually supplemented with facts about the effects of different ranges of calories intake in order to promote adherence to a low-calorie diet for overweight patients. In our case, we were thus interested in assessing how objective knowledge about COVID-19 is related to perceived efficacy of the actions taken by the governments, peers and their own to counteract the effects of the pandemic. Conversely, we reasoned that belief in conspiracy theories about COVID-19 could also influence to some extent the perceived efficacy of actions to contain the pandemic [35]. For instance, if participants believe that installing 5G mobile antennas are the main reason for governmental lockdown decisions, then they may perceive lockdown as unnecessary or ineffective [36]. Notably, trust, beliefs in efficacy and feelings of personal risk (including fear of contagion and death) contribute to compliance with recommended preventive health behaviours [23], like the issued restrictions for COVID-19. In our study, we examined the factors associated with the aforementioned predictors.

## 1.3. Multinational sample

Importantly for the purpose of the current study, risk perception and risk preferences can vary by cross-national differences [37–39]. Intensifying these differences, political leaders around the world had distinct responses to the outbreak since it was first identified in China. These differences may have affected public trust in those leaders, as well as the responses of individuals to the pandemic [24,40]. For instance, and in contrast with Norway and Germany leaders, the Brazilian president Bolsonaro criticized the quarantine measures imposed by the state governors [41]. After his public statement, there has been an increase in Brazilians leaving their houses and breaking the quarantine, indicating reduced belief in the efficacy of these measures [42]. Here, we investigated whether country of residence influenced how people perceived the efficacy of their own, other and governmental reactions, as well as of reactions and restrictions in general. We did not put forward any specific hypothesis regarding the influence of cultural or even regional factors, given that our study was exploratory and took advantage of convenience and opportunistic sampling. Nevertheless, our results can provide initial evidence for cross-national similarities and differences in people's reactions to the pandemics.

Our sample included participants from six countries. Our only criteria of inclusion were the interest of researchers in conducting the study. At the time we collected the data, these countries varied in the degree of restrictions they had adopted (figure 1). The restrictions varied from very restrictive (e.g. lockdowns in Israel as of 22 March 2020) to less restrictive (e.g. Norway, Germany and Brazil, where schools were closed, home office was recommended but people could still leave the house without any legal repercussions).

To provide context for our cross-national comparisons, table 1 provides the reported number of positively tested COVID-19 cases in mid-March and end of March, both in absolute numbers and per 1 million inhabitants per country, as well as confirmed COVID-19 deaths. The confirmed COVID-19 cases per million people depend heavily on the testing regime of the country and should be treated with caution.

**Figure 1.** Country-wise restrictions stringency index for the data collection period, based on [43] for Brazil, Colombia, Germany, Norway, Israel and USA. Start date and end date of the survey in the various languages indicated by arrows. Graph modified from ourworldindata.org [44].

**Table 1.** Relative and absolute number of confirmed COVID-19 cases and deaths as of 12 and 30 March 2020 for the six countries, data retrieved from ourworldindata.org. The survey's language and the authors targeted often a specific country, e.g. Brazil, but respondents from other countries could partake, as well as some Brazilians could partake in the English survey. We therefore report the confirmed cases from the launch of the English survey (12 March) until 30 March.

| country | confirmed cases on 12 Mar 2020 | confirmed cases on 30 Mar 2020 | confirmed cases per million on 30 Mar 2020 | factor change between 12 Mar and 30 Mar | confirmed deaths on 12 Mar 2020 | confirmed deaths on 30 Mar 2020 | confirmed deaths per million on 30 Mar 2020 |
|---|---|---|---|---|---|---|---|
| Brazil | 52 | 4256 | 20.02 | ∼82 | 0 | 136 | 0.64 |
| Colombia | 9 | 702 | 13.8 | ∼78 | 0 | 10 | 0.2 |
| Germany | 1567 | 57 298 | 683.88 | ∼37 | 3 | 455 | 5.43 |
| Israel | 82 | 4247 | 490.67 | ∼52 | 0 | 15 | 1.73 |
| Norway | 489 | 4102 | 756.65 | ∼8 | 0 | 22 | 4.06 |
| USA | 1312 | 143 025 | 432.1 | ∼109 | 30 | 2509 | 7.58 |

## 1.4. Hypotheses

Taken together, behavioural, cultural and psychological factors can impact individuals' responses to pandemics and to the policies to contain them [4,31,45,46]. In this multinational observational study, we measured how much respondents were affected by their countries' restrictions shortly after those restrictions were announced, as well as respondents' perceived risk of contracting COVID-19, perceived effectiveness of their own, others or governmental reactions and perceived effectiveness of a range of restrictions, e.g. school closings. We also assessed knowledge about COVID-19, feelings of paranoia, feeling of controlling the outbreak, as well as general distress, as we can expect them to

increase the more stringent and sudden the objective restrictions were put in place and the harsher they were subjectively perceived.

Based on the reviewed literature, we formed the following predictions (see pre-registration at https://osf.io/bh2cz/). **H1, perceived severity:** Firstly, we predicted that personal life- and work-style changes will be rated as the measures most severely affecting daily life, and participants who experience these restrictions will perceive them as more severe than those who did not (yet) experience them [15,17].

**H2, perceived efficacy of reactions:** Secondly, we hypothesized that perceived efficacy of own, other, and governmental reactions will be related to perceived risk of COVID-19 contagion, knowledge about the virus, feeling of controlling the outbreak, numbers of protective actions taken, worry/fear about COVID-19, and general distress [18,24,45]. In a follow-up analysis, we included country as a predictor.

**H3, perceived efficacy of restrictions:** Thirdly, we hypothesized that the perceived efficacy of restrictions will be correlated with perceived risk of COVID-19 contagion, numbers of protective actions taken and country [32,47].

**H4, self-rated mental health:** Fourthly, we hypothesized that general distress will be associated with feeling of controlling the outbreak, perceived risk of COVID-19 contagion, number of protective actions taken, knowledge and worry/fear about COVID-19 [10,14,23,48].

Across analyses, we included gender and age as covariates to account for possible differences in e.g. health risk perception [49]. After pre-registering our analysis, we learned about the stringency index [43], a measure to quantify institutional responses to COVID-19 based on nine indicators of government responses (e.g. school closings, border closings). To account for potential variability accounted for by changes in stringency during the duration of data collection, we ran exploratory analyses using the stringency index for each country as it tracked development during the 10–14 days the survey was open.

# 2. Methods

We designed a survey in eight different languages: Arabic, Brazilian Portuguese, English, German, Hebrew, Italian, Latin-American Spanish and Norwegian. The English, German and Norwegian survey was distributed online on 12 March and closed on 25 March. The survey opened in Israel (Hebrew) on 16 March, closed 25 March, opened in Israel (Arabic) and in South America on 20 March, closed on 30 March (08.00 UTC). The survey was distributed on social media websites and messaging programs using a snowball sampling method. All participants provided their informed consent in a manner approved by the relevant ethics board (see Ethics section).

The survey was open during a phase where particularly Colombia and Israel transitioned from partial to a more complete lockdown, but all six countries progressed to more strict government policies during data acquisition (figure 1).

## 2.1. Variables and indices

We measured several distinct constructs in our survey, across four categories: restrictions and reactions, psychological measures, paranoia and knowledge measures, and general demographic measures. We describe the measures for each category below.

## 2.2. COVID-19 restrictions and reactions

### 2.2.1. Experienced restrictions

We asked what effect the outbreak had on the respondent: being quarantined, under a travel ban, closing of schools and universities, mandatory home office, cancellation of culture and sport events, transport restriction or a family member being infected. Not being affected by any restriction was also an answer option. Having contracted the virus or recovered from it was included in the Spanish and Brazil–Portuguese survey. We measured the experienced restrictions on a nominal scale, with multiple answers possible per participant.

### 2.2.2. Impact of restrictions

We asked how impactful the restrictions asked in the previous question (see 'experienced restrictions') were for the respondents' daily life. For each restriction, we used a scale from 1 = not affecting my daily life much, to 3 = will affect my daily life a lot.

### 2.2.3. Efficacy of restrictions

We asked about the perceived effectiveness of six specific restrictions in reducing the outbreak (avoiding social gatherings, cancellation of meetings/culture/sport events, closing school/kindergarten/university, closing transport (airports, trains, buses, ferries), travel ban for one to two months, quarantine at home). The scale ranged from 0 = not effective at all to 100 = most/absolutely effective. We calculated three scores for efficacy of restrictions: (i) efficacy of school/kindergarten/university closings; (ii) efficacy of quarantine; (iii) efficacy of public life restrictions, i.e. average score for efficacy of transport closings, travel ban, social gatherings and cancelling culture and sport events.

### 2.2.4. Protective actions

We asked which protective actions the participants were taking, including hand washing, social distancing, online meetings and cancelling travel. In the Spanish and Brazilian Portuguese survey, we also asked for stockpiling: for food, household items and medicine, respectively. In our analyses, we used the number of protective actions performed.

### 2.2.5. Perceived efficacy of reactions

We measured the perceived efficacy of own, others' and governmental actions in general terms. Example item for own: 'My actions are effective in limiting the outbreak'. These three items were measured on a visual analogue scale (VAS) from 0 to 100. We also calculated an average score across the three items. Efficacy of actions is one of the crucial components of fear appeal in Rogers' protection motivation theory [30].

### 2.2.6. Satisfaction with governmental reactions

We also measured whether respondents believe their country is taking enough actions to fight the outbreak on a scale of 1 = yes, 2 = do not know and 3 = no.

## 2.3. COVID-19 psychological measures: feeling control of the outbreak, perceived risk, worry and fear about contagion, and general distress

### 2.3.1. Feeling of controlling the outbreak

We measured the feeling of control with the single item 'I feel we can control the outbreak of the Coronavirus' on a VAS from 0 to 100.

### 2.3.2. Worry and fear

We measured worry and fear for COVID-19 with two items on a VAS from 0 to 100. The two items were: 'I am very worried about the Corona virus outbreak' and 'I am scared of the Coronavirus outbreak'. We used the average score across the two items. Worry and fear addresses the perceived magnitude of noxiousness of COVID-19 contagion [30].

### 2.3.3. Perceived risk of COVID-19

We included three items to ask about (i) risk of contracting COVID-19 within the next week, (ii) within the next two months, and (iii) getting seriously ill if contracted. We used a VAS scale from 0 (no risk) to 100 (certainty) and calculated an average score across the three items as perceived risk about COVID-19. Perceived risk is a proxy for the subjective probability of COVID-19 contagion [30].

### 2.3.4. CORE-9

To assess general distress, we used the **C**linical **O**utcomes in **R**outine **E**valuation 10 item short version [50], with all but the 'plan to end my life' item (henceforth CORE-9). The CORE measures daily life functions, symptoms and problems, and well-being. Example items are; 'I have felt able to cope when things go wrong' and 'Talking to people has felt too much for me'. Analysis is based on the average score of those nine items. Internal consistency of this scale was McDonalds $\Omega = 0.789$. The CORE is

not a diagnostic tool but [50] applied a cut-off between clinical and non-clinical samples at 1.0/1.1 for general distress.

## 2.4. COVID-19 paranoia and knowledge measures

### 2.4.1. CAPE-P items

To measure paranoia, we included five paranoia items, three anomalous perception items and two grandiosity items, rated from 1 = never to 4 = nearly always, from the Community Assessment of Psychotic-like Experiences questionnaire (CAPE, [51]). We used the average score from the 10 items in the analysis. Internal consistency of the scale was McDonalds $\Omega = 0.788$. The CAPE is not a diagnostic tool; however, the CAPE-P with a cut-off of 1.47 had a positive predictive value of 66% to detect people at ultra-high risk for psychosis [52].

### 2.4.2. CAPE-C items

We included three control items (aliens, famous historical person, mental collapse) as quality checks [53]. We used a score higher than 8 as an exclusion criterion.

### 2.4.3. Knowledge score

We presented three items asking how much respondents endorse different conspiracy theories such as 'The virus is part of a Chinese biological weapons program'. We presented four statements varying in their factual truth, e.g. 'The virus belongs to the SARS family'. Responses were scored on a VAS from 0 = not true at all to 100 = absolutely true. We calculated a difference score between belief in conspiracy theories (average of three items) and knowledge (average of four items). A negative score indicates endorsement of conspiracy theories about the virus. Internal consistency was McDonalds $\Omega = 0.671$.[1]

## 2.5. General demographic measures

We also asked for age in years, gender (male = 0, female = 1, and response option 'other' coded as 2), and country of residency. Local versions (Israel, South America) included a few additional demographic variables that were not included in our analyses.

No response apart from consent was forced. The survey was implemented in Qualtrics.

## 2.6. Inclusion/exclusion criteria

For all analyses, we included individuals who were at least 18 years old. We excluded all participants scoring more than eight points on the three CAPE quality control items (score can range from 3 to 12). We expected missing data as we did not force all choices. Answering fewer than 70% of the items on a scale led to its exclusion for the analysis requiring this scale. Participants with more than 50% missing data in total were excluded. We also excluded invalid responses by excluding participants that answered the survey in less than 3 min, an impossible task given human reading speeds.

## 2.7. Analyses

To address our hypothesis 1, we cross-tabulated the type of experienced restrictions on daily life with the impact these restrictions had on daily life. We also compared the respondents who reported being affected by the outbreak to those who reported not being affected by asking the latter to rate how such restrictions would impact them.

To address our hypothesis 2, we performed a multiple linear regression analysis for perceived efficacy of reactions as an outcome variable. Predictor variables were: knowledge score, feeling of controlling the outbreak, perceived risk, number of actions taken, general distress, worry/fear about COVID-19, gender and age. We did not include the efficacy of restrictions in this analysis, as the perceived efficacy of

---

[1]If dropping the item 'COVID-19 belongs to SARS family of viruses' the scale's internal consistency improved to be 0.713. However, this item is 100% true, whereas the other items are not 100% true or false, just extremely unlikely or most probably true. We therefore did not remove this item from the scale.

reactions score, although general in its nature, can be impacted by the impression of specific restrictions enforced by the government. To assess the influence of lockdown severity, we followed up with two exploratory analyses: (i) using the stringency index as another predictor in the regression model, and (ii) a GLM where country (six countries for which we had at least 50 respondents) was a between-groups factor and all other variables were entered as covariates.

To address our hypothesis 3, we computed a GLM for the efficacy of restrictions (closing schools, quarantine, public life) as the outcome variables and perceived risk, numbers of actions taken, gender and age as covariates. Country was entered as a between-groups factor.

To address our hypothesis 4, we used a multiple linear regression analysis for general distress (CORE-9 score) as an outcome variable, and the following variables as predictors: feeling of control, perceived risk, number of actions taken, paranoia (CAPE-P score), knowledge,[2] worry/fear of COVID-19, gender and age.

Data analysis was performed in R and JASP v. 11.1.0. [54].

# 3. Results

There were 2285 participants after we removed those who completed less than 50% and/or answered faster than 3 min (removal of $n = 1203$). Eight participants had a CAPE-C score of 8 or higher and were excluded from all analysis. Another 16 participants were not yet 18 years old (nearly all were 17 years old, partially overlapping with the participants excluded by the CAPE-C score). That left a total of 2264 valid participants. Three hundred and thirty-six did not indicate their country of residence and 101 resided in countries for which we obtained less than 50 respondents. Accordingly, analyses by country had maximally 1822 respondents. Analysis using the stringency index had $N = 1893$, as there was no index value for some countries [43]. Note the number of participants was often smaller as not all answered all items and we did not impute missing values.

## 3.1. Descriptives

The average age of our respondents was 33.0 years (s.d. = 12.5, range 18–81). Eleven indicated 'other' as gender, 1497 were female, 715 were male, and 49 did not indicate their gender. Every country with more than 40 respondents had more female than male respondents. For the six countries with at least 50 respondents the demographics are presented in table 2.

The majority of our participants responded with 'their country was doing enough to fight the outbreak' ($n = 899$). Seven hundred and seventy participants said 'their country was not doing enough' and 588 indicated they 'didn't know whether their country was doing enough to fight the outbreak'.

As for protective actions, 2204 out of 2264 (97.5%) respondents washed their hands, 2020 (89.4%) engaged in spatial (social) distancing, 1219 (54%) in online meetings and 1084 (48%) cancelled travel. On the other hand, 235 (10.4%) indicated that they did not perform any actions to protect themselves from the virus, and 201 (9.2%) answered that they did not perform any action to protect others. Of these participants, a total of 111 participants were doing neither. Among the 235 saying they did not perform any protective actions for themselves 210 did engage in hand washing and 133 in social distancing. Similarly, among the 201 saying they did not perform any actions to protect others, 182 engaged in hand washing and 119 in social distancing (statistical tests in electronic supplementary material, see also figure S1). Among South Americans 51.9% stockpiled food, 38% stockpiled household items and 17.8% stockpiled medicine. Table 3 summarizes the psychological factors across six countries as well as among all those from countries with less than 50 participants and those that did not indicate their country of residency.

Feeling of controlling the outbreak was lowest in Germany, whereas respondents from Brazil, Colombia and Israel reported up to 19 points higher feelings of controlling the outbreak. Perceived risk was below 50 in all countries, and also lower than worry or fear about COVID-19 contagion. Categorizing distress severity [50], participants from Colombia had on average moderate levels of distress (1.5–2.0); participants from Brazil and the USA reported on average mild levels of distress (1.1–1.5); participants from Israel, Norway and Germany had on average low levels of distress. In all countries, participants endorsed the factual more than the conspiracy items. Regarding paranoia, participants from Brazil, Colombia and Israel scored slightly above the cut-off point.

---

[2]In the pre-registration, we forgot to include knowledge as predictor in this analysis.

**Table 2.** Age, gender and stating being affected by the outbreak in six countries. Other countries: Sweden: 18; Canada: 18; The Netherlands: 15; UK: 14; France: 8; Finland: 6; Austria: 4; Belgium, Lithuania, Turkey: 2 each.

| country | Brazil | Colombia | Germany | Israel | Norway | USA |
|---|---|---|---|---|---|---|
| N | 204 | 418 | 99 | 271 | 732 | 50 |
| mean age | 33.95 | 24 | 39.1 | 33.5 | 36.2 | 34.66 |
| male/female/other | 56/147/1 | 151/266/4 | 33/66/0 | 93/196/0 | 205/527/1 | 13/37/0 |
| affected yes/no | 199/5 | 413/13 | 41/58 | 253/39 | 557/180 | 45/5 |

Regarding differences by country, we found that all outcomes (feeling of control, perceived risk, worry/fear, general distress, knowledge and paranoia) differed between the countries, all $p < 0.001$ with effect sizes ranging from $\eta^2 = 0.031$ to 0.221 (see the electronic supplementary material).

Next, we assessed the relationship between efficacy of actions and efficacy of restrictions. The perceived efficacy of own, others' or governmental actions did not correlate highly with the efficacy of the restrictions of school closings, quarantine and public life (figure 2). The three categories of restrictions were all rated as highly efficacious (see also results for hypothesis 3, and electronic supplementary material, figure S2 per country).

## 3.2. Hypothesis 1: direct impact of restrictions on daily life

Hypothesis 1 regarding perceived severity was confirmed. Restricting analysis to those who self-reported to be affected ($n = 1804$), we tested how they were affected (each respondent could provide multiple answers), and how severely their daily lives were affected on a scale from not much, somewhat, to very much (figure 3a). One respondent had recovered from COVID-19 and none of our participants had COVID-19, but 222 (9.8%) stated that a family member was infected or ill. One thousand and ninety-three (48.4%) had to take home office, 906 (40.1%) were affected by cancellation of meetings, sport and or culture events, 849 (37.6%) by having to keep social distance, 686 (30.3%) by travel ban, 601 (26.6%) by schools/kindergarten/universities being closed, and 630 (27.9%) by being in quarantine.

Participants were severely (71%) affected by school (kindergarten, university) closings (figure 3a). In comparison, social distancing, cancellation of sport and cultural events and reduction in transport were not rated as affecting daily life severely. Nearly half of those affected by quarantine at home replied that it did not affect their daily life very much, only somewhat.

Next, we compared those severity percentages to those of people who were not (yet) affected. We calculated a difference of perceived severity among those affected and those not affected and how they anticipated those restrictions would impact their daily life. As can be seen from figure 3b, among those that were not currently affected by, e.g. school closings, they anticipated that this action would affect them less in the coming weeks compared to respondents already being impacted. For all six restrictions, those who were already experiencing the restrictions, rated them as more severely affecting their daily life compared to those who were not yet experiencing it. The disparity between predicted and experienced effect was smallest for quarantine.

Finally, participants could also choose the answer option 'other': 345 participants provided written responses to describe how the situation affected them. Many mentioned economic concerns or having lost their job. A few were also concerned about their future as they could not go to university and graduate. Some also mentioned depression, being anxious about relatives working in the health sector or that their relatives (siblings, parents, spouse) were in poor health (cancer, brain tumour, heart operations, etc.) and may not survive a COVID-19 infection.

## 3.3. Hypothesis 2: How do participants perceive efficacy of their own, others' and governmental reactions?

Perceived efficacy of reactions to limit the outbreak (i.e. the average score of the efficacy of own, others' and governmental reactions) had a mean of 70 (s.d. = 18.8). Perceived efficacy was rated highest for own reactions ($M = 75.07$, s.d. = 21.59), followed by similar ratings of perceived efficacy for the reactions of others ($M = 67.5$, s.d. = 24.97) and of the government ($M = 67.45$, s.d. = 25.61). This difference between agents of the reaction was significant, $F_{1.984,2896.442} = 57.5$, $p < 0.001$, $\eta^2 = 0.014$. The perceived efficacy of

**Table 3.** Descriptive statistics for knowledge, feeling of control, perceived risk, distress and paranoia. Feeling of controlling ranged in all countries from 0 to 100.

| | | Brazil | Colombia | Germany | Israel | Norway | USA | other | unknown |
|---|---|---|---|---|---|---|---|---|---|
| feeling of controlling the outbreak | mean | 57.43 | 59.77 | 40.32 | 57.41 | 51 | 48.84 | 50.38 | 52.21 |
| | s.d. | 28.99 | 24.93 | 27.12 | 27.22 | 26.01 | 27.70 | 29.04 | 27.62 |
| worry/fear about COVID-19 | mean | 79.27 | 63.47 | 46.38 | 61.46 | 50.19 | 66.98 | 49.37 | 54.85 |
| | s.d. | 21.96 | 26.14 | 25.25 | 27.35 | 26.61 | 24.79 | 28.10 | 28.16 |
| | min | 3.5 | 0 | 1 | 0 | 0 | 10 | 0 | 0 |
| | max | 100 | 100 | 92 | 100 | 100 | 100 | 100 | 100 |
| perceived risk about contagion | mean | 43.92 | 33.5 | 40.14 | 39.28 | 41.99 | 46.91 | 41.22 | 40 |
| | s.d. | 22.72 | 20.81 | 19.32 | 20.22 | 20.58 | 22.2 | 21.13 | 20.87 |
| | min | 0 | 0 | 1 | 0 | 0 | 8.33 | 0 | 0 |
| | max | 94 | 93.67 | 79.67 | 100 | 100 | 91.67 | 93.33 | 100 |
| general distress | mean | 1.25 | 1.55 | 0.84 | 1.1 | 0.76 | 1.21 | 1.06 | 0.78 |
| | s.d. | 0.78 | 0.76 | 0.58 | 0.62 | 0.55 | 0.87 | 0.74 | 0.61 |
| | min | 0 | 0 | 0 | 0 | 0 | 0.11 | 0 | 0 |
| | max | 3.56 | 3.78 | 2.44 | 3.00 | 3.22 | 3.22 | 3.44 | 3.33 |
| paranoia | mean | 1.53 | 1.7 | 1.43 | 1.51 | 1.29 | 1.41 | 1.41 | 1.33 |
| | s.d. | 0.36 | 0.39 | 0.26 | 0.31 | 0.22 | 0.28 | 0.29 | 0.25 |
| | min | 1 | 1 | 1 | 1 | 1 | 1 | 1 | 1 |
| | max | 3 | 3.5 | 2.2 | 3.5 | 2.5 | 2.4 | 2.4 | 2.6 |
| knowledge score | mean | 38.25 | 18.86 | 40.84 | 34.3 | 35.2 | 37.82 | 39.43 | 40.47 |
| | s.d. | 22.54 | 28.25 | 22 | 25.41 | 20.87 | 19.67 | 25.03 | 20.07 |
| | min | −38 | −75 | −35 | −51 | −70 | −17 | −34 | −33 |
| | max | 93 | 99 | 88 | 83 | 100 | 70 | 93 | 100 |

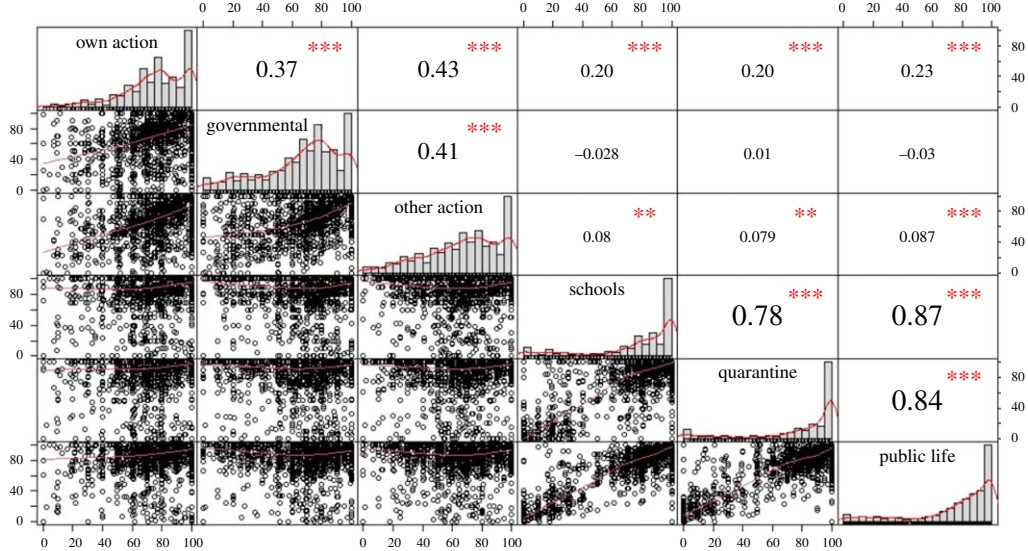

**Figure 2.** Correlation plot for the perceived efficacy of own, others' or governmental reactions and efficacy of restrictions such as school closings, quarantine and public life restrictions (for a split by country, see electronic supplementary material, figure S2).

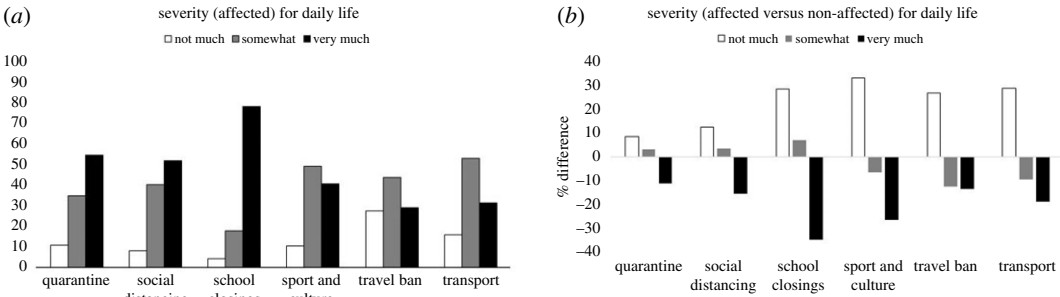

**Figure 3.** (*a*) Among those affected by COVID-19, the severity of impact on daily life was highest for school/university closings. (*b*) Difference in percentage in rating the severity of six restrictions. Positive score: non-affected rated it higher, negative score: higher rating among those who were affected. All values in percentage.

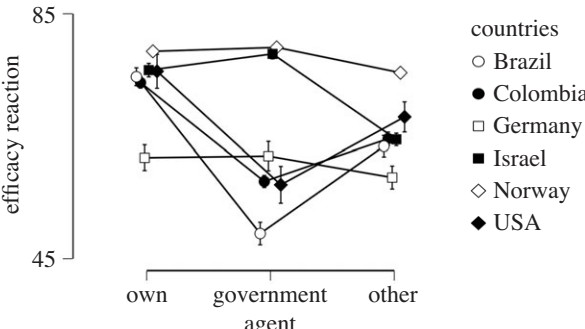

**Figure 4.** Perceived efficacy of own, governmental or others' reactions on limiting the outbreak by country.

own, others' or governmental reactions differed by country, $F_{5,1460} = 36.656$, $p < .001$, $\eta^2 = 0.112$ and also yielded a significant interaction, $F_{9.919,2896.442} = 32.035$, $p < 0.001$, $\eta^2 = 0.039$. Perceived efficacy of reactions was lowest in Germany ($M = 61.52$, s.d. $= 20.44$), followed by Brazil ($M = 62.83$, s.d. $= 21.38$), Colombia ($M = 65.26$, s.d. $= 18.18$), the USA ($M = 66.39$, s.d. $= 18.03$), Israel ($M = 72.69$, s.d. $= 17.11$) and Norway ($M = 77.8$, s.d. $= 15.65$). As can be seen in figure 4, Brazil, Colombia and the USA rated governmental reactions as less efficient than their own reactions or that of others in limiting the COVID-19 outbreak (post-hoc tests in electronic supplementary material).

**Table 4.** Perceived efficacy of the average of own, others' and governmental reactions.

| | unstandardized | s.e. | standardized | $t$ | $p$ | 95% CI lower | upper |
|---|---|---|---|---|---|---|---|
| (intercept) | 56.855 | 3.428 | | 16.586 | <0.001 | 50.132 | 63.578 |
| knowledge | 0.029 | 0.017 | 0.039 | 1.675 | 0.094 | −0.005 | 0.063 |
| feeling of controlling the outbreak | 0.211 | 0.016 | 0.305 | 13.196 | <0.001 | 0.180 | 0.242 |
| perceived risk | 0.071 | 0.021 | 0.082 | 3.398 | <0.001 | 0.030 | 0.112 |
| number protective actions | 2.330 | 0.518 | 0.104 | 4.502 | <0.001 | 1.315 | 3.345 |
| general distress | −3.425 | 0.719 | −0.136 | −4.766 | <0.001 | −4.835 | −2.015 |
| worry/fear for COVID-19 | 0.032 | 0.018 | 0.047 | 1.781 | 0.075 | −0.003 | 0.067 |
| gender | 2.918 | 0.936 | 0.073 | 3.116 | 0.002 | 1.081 | 4.754 |
| paranoia | −6.843 | 1.463 | −0.126 | −4.678 | <0.001 | −9.712 | −3.974 |
| age | 0.026 | 0.038 | 0.017 | 0.679 | 0.497 | −0.048 | 0.099 |

For the main analysis testing Hypothesis 2, we used the average score of the efficacy of reaction rating. In line with our hypothesis, nearly all predictors contributed significantly to the average perceived efficacy of reactions, and explained 17% of the variance, $F_{8,1626} = 36.757$, $p < 0.001$. The strongest predictor was the feeling of controlling the outbreak. The larger the feeling of controlling the outbreak the more efficient the reactions were perceived ($\beta = 0.305$). Similarly, the larger the perceived risk and the more protective actions performed, the higher was the perceived efficacy of the reactions. A low score on paranoia and low general distress also contribute to higher perceived efficacy of reactions. Age, knowledge about the coronavirus and worry/fear about COVID-19 did not significantly contribute to the perceived efficacy of reactions (table 4). Predictors did not exhibit collinearity, all variance inflation factors (VIF) were below 1.6. Using this as the null model and adding the stringency index, we found that the model improved by only 0.1% (see the electronic supplementary material).

To assess the influence of country, we fitted a similar GLM with country added as between-groups factor (table 5). Notably, unlike the previous analysis, perceived risk had no statistically significant effect, but worry/fear had. Feeling of controlling the outbreak and number of protective actions (hand washing, social distancing, etc.) remained positively associated with the perceived effectiveness of own, others' and governmental reactions.

Comparing tables 4 and 5 (see also figure 6), feeling of controlling the outbreak was consistently a strong predictor for the efficacy of own, others' or government's reactions to limit the outbreak. Interestingly, the second largest predictor when including between-country differences in the model, other than these country-level differences, was worry and fear about contagion. However, without the country as between-factor, perceived risk about contagion explained a significant portion of the variance. Thus, country differences underlie here whether either worry/fear or personal risk of contagion contribute to participants' perception of efficacy of reactions. Regardless, across both analyses a heightened feeling of threat (either by scoring high on perceived risk or high on worry/fear) contributed to a higher perceived efficacy of reactions score.

We conducted three additional exploratory (not pre-registered) analyses. First, we looked at whether perceived efficacy of reactions differed among those that stated that their country did enough or did not do enough to fight the outbreak. 'Do not know' responses were excluded in this analysis. A logistic regression showed that respondents who rated the risk of contracting COVID-19 lower and were less worried were more likely to state that their country did enough. In addition, stating their country did enough was also positively predicted by age, by rating the efficacy of governmental reactions to limit the outbreak as high and by rating the efficacy of own reactions as low (table 6).

Second, we compared the perceived efficacy of reactions between the six countries. There was a main effect for who performs the reaction (own, government or others): $F_{1.979,2862.305} = 23.356$, $p < 0.001$, $\eta^2 = 0.006$. There was a main effect for country: $F_{5,1446} = 25.323$, $p < 0.001$, $\eta^2 = 0.079$, and a main effect for satisfaction (yes, does enough; no, does not enough; do not know): $F_{2,1446} = 10.506$, $p < 0.001$,

**Table 5.** Perceived efficacy of own, others' and governmental reactions controlling for country-level differences. Type III Sum of Squares.

| cases | sum of squares | d.f. | mean square | F | p | $\eta^2$ |
|---|---|---|---|---|---|---|
| countries | 39 646.99 | 5 | 7929.4 | 29.15 | <0.001 | 0.081 |
| knowledge | 397.37 | 1 | 397.37 | 1.46 | 0.227 | 0.001 |
| feeling of controlling the outbreak | 49 048.02 | 1 | 49 048.02 | 180.28 | <0.001 | 0.100 |
| perceived risk | 331.6 | 1 | 331.6 | 1.22 | 0.270 | 0.001 |
| number of protective actions | 3290.08 | 1 | 3290.08 | 12.09 | <0.001 | 0.007 |
| general distress | 2504.7 | 1 | 2504.7 | 9.21 | 0.002 | 0.005 |
| worry/fear | 4329.57 | 1 | 4329.56 | 15.91 | <0.001 | 0.009 |
| gender | 964.66 | 1 | 964.66 | 3.55 | 0.060 | 0.002 |
| paranoia | 1282.33 | 1 | 1282.33 | 4.71 | 0.030 | 0.003 |
| age | 52.45 | 1 | 52.44 | 0.19 | 0.661 | 0.000 |
| residual | 387 157.96 | 1423 | 272.07 | | | |

$\eta^2 = 0.013$. The interaction between who performs the reaction and country was significant: $F_{9.897,2862.305} = 15.207$, $p < 0.001$, $\eta^2 = 0.019$, as well as the interaction between reaction and satisfaction: $F_{3.959,2862.305} = 17.093$, $p < 0.001$, $\eta^2 = 0.009$. As can be seen in figure 5, own reactions were generally rated as effective but perceived efficacy of governmental reactions differed by degree of satisfaction with how countries fought the outbreak. More than 50% were satisfied in Israel and Norway, whereas the majority of participants from Brazil and the USA were not satisfied with how their country was fighting the outbreak.

Thirdly, we ran the model of hypothesis 2 separately for own, others' and governmental reaction. Efficacy of own reactions might relate to self-efficacy [23], whereas the reactions of others and the government can be related to the more general efficacy of a protective response. It could be the case that high 'self-efficacy' is positively related to one's worry/fear whereas 'governmental efficacy' might not be positively related to one's worry/fear.[3]

Feeling of controlling the outbreak and efficacy of own reaction limiting the outbreak were positively but not highly correlated, $\rho = 0.26$, 95% CI [0.215; 0.303], explaining only 7% of the variance. Accordingly, we ran the full model as for hypothesis 2. The predictors for perceived efficacy of own reactions limiting the outbreak explained 14.3% of the variance. Significant predictors were feeling of controlling the outbreak ($\beta = 0.269$), number of protective actions ($\beta = 0.149$), general distress ($\beta = -0.11$) and worry/fear ($\beta = 0.178$), all with $p < 0.001$, as well as paranoia ($\beta = -0.056$, with $p = 0.044$). The lower a person's general distress was and the more the person worried about contagion the more efficacious own reactions were rated.

The predictors for perceived efficacy of others' reactions limiting the outbreak explained only 9.3% of the variance. Significant predictors were feeling of controlling the outbreak ($\beta = 0.219$), paranoia ($\beta = -0.135$), perceived risk ($\beta = 0.086$), all three with $p < 0.001$ and general distress ($\beta = -0.078$, $p = 0.009$).

The predictors for perceived efficacy of governmental reactions limiting the outbreak explained 12.9% of the variance. Significant predictors were feeling of controlling the outbreak ($\beta = 0.233$), general distress ($\beta = -0.121$), gender ($\beta = 0.105$), perceived risk ($\beta = 0.088$), paranoia ($\beta = -0.1$), all five $p$s < 0.001, worry/fear ($\beta = -0.08$, $p = 0.004$) and number of protective actions ($\beta = 0.06$, $p = 0.012$). In contrast to perceived efficacy of own reaction, less worry/fear was related with higher efficacy scores for governmental reactions. Paranoia was a larger predictor for perception of the efficacy of governmental and other reactions than own reactions. Regardless, across own, others' and governmental reactions, the largest predictor for perceived efficacy of reactions was feeling of controlling the outbreak. The higher this feeling the more efficacious all reactions were perceived (figure 6).

[3]We thank an anonymous reviewer for this suggestion.

**Table 6.** Coefficients of the logistic regression, testing the effects of the listed predictors on whether one thinks their country is not doing enough to fight the outbreak. Country_does_enough level 'not enough' coded as class 1.

| | estimate | standard error | odds ratio | z | Wald test | | | | 95% confidence interval | |
| | | | | | Wald statistic | d.f. | p | | lower bound | upper bound |
|---|---|---|---|---|---|---|---|---|---|---|
| (intercept) | 0.910 | 0.647 | 2.5 | 1.407 | 1.98 | 1 | 0.159 | | −0.357 | 2.177 |
| stringency index | −0.001 | 0.005 | 1 | −0.229 | 0.05 | 1 | 0.818 | | −0.011 | 0.008 |
| knowledge | 0.003 | 0.003 | 1 | 1.089 | 1.19 | 1 | 0.276 | | −0.002 | 0.009 |
| feeling control | −0.002 | 0.003 | 1 | −0.837 | 0.7 | 1 | 0.402 | | −0.008 | 0.003 |
| perceived risk | 0.016 | 0.004 | 1.02 | 4.435 | 19.67 | 1 | <0.001 | | 0.009 | 0.023 |
| general distress | 0.124 | 0.123 | 1.13 | 1.002 | 1 | 1 | 0.316 | | −0.118 | 0.365 |
| worry/fear | 0.014 | 0.003 | 1.01 | 4.450 | 19.8 | 1 | <0.001 | | 0.008 | 0.020 |
| gender | −0.169 | 0.157 | 0.84 | −1.074 | 1.15 | 1 | 0.283 | | −0.478 | 0.139 |
| paranoia | 0.095 | 0.245 | 1.1 | 0.387 | 0.15 | 1 | 0.699 | | −0.386 | 0.575 |
| own reaction | 0.016 | 0.004 | 1.02 | 3.584 | 12.84 | 1 | <0.001 | | 0.007 | 0.024 |
| government reaction | −0.052 | 0.004 | 0.95 | −12.741 | 162.32 | 1 | <0.001 | | −0.060 | −0.044 |
| other reaction | 0.007 | 0.003 | 1.01 | 1.877 | 3.53 | 1 | 0.060 | | −0.000 | 0.013 |
| age | −0.015 | 0.006 | 0.99 | −2.356 | 5.55 | 1 | 0.018 | | −0.028 | −0.003 |

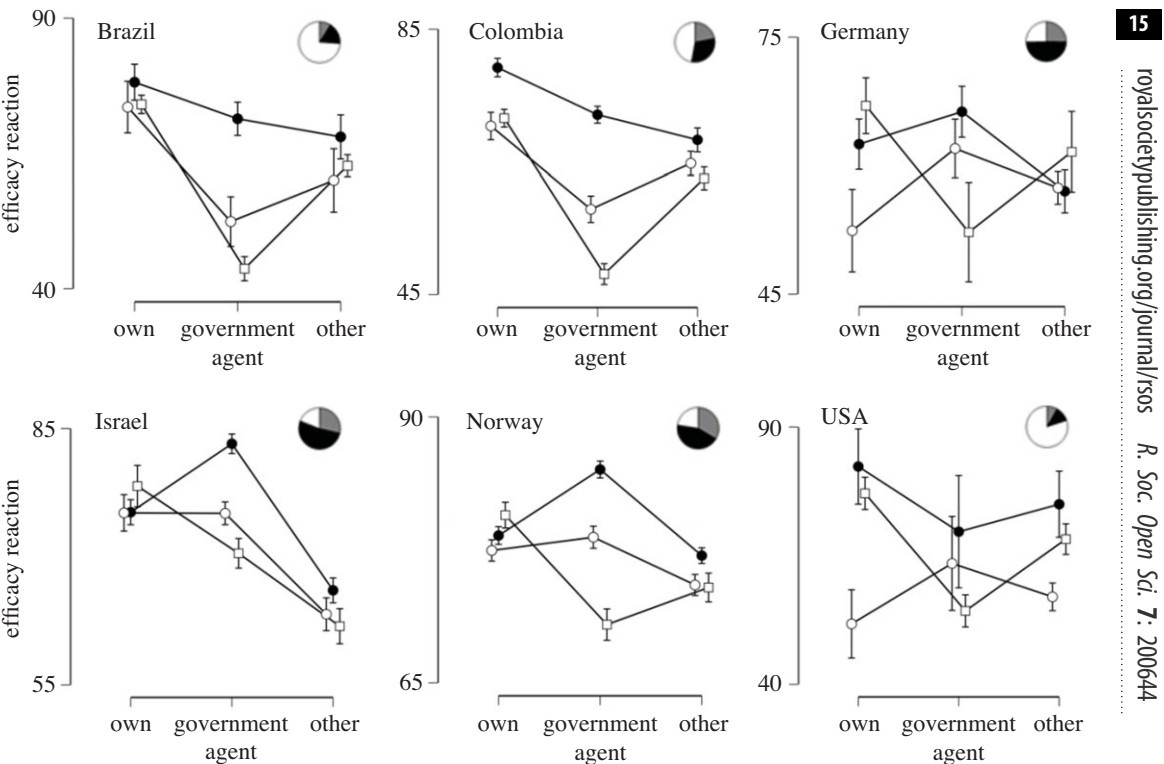

**Figure 5.** Perceived efficacy of reactions per country and action. Filled circles and black in the pie chart inserts: participants were satisfied with how their country fought the outbreak; white circles and grey in pie chart inserts: participants stating do not know whether their country does enough; white squares and white pie: participants were not satisfied.

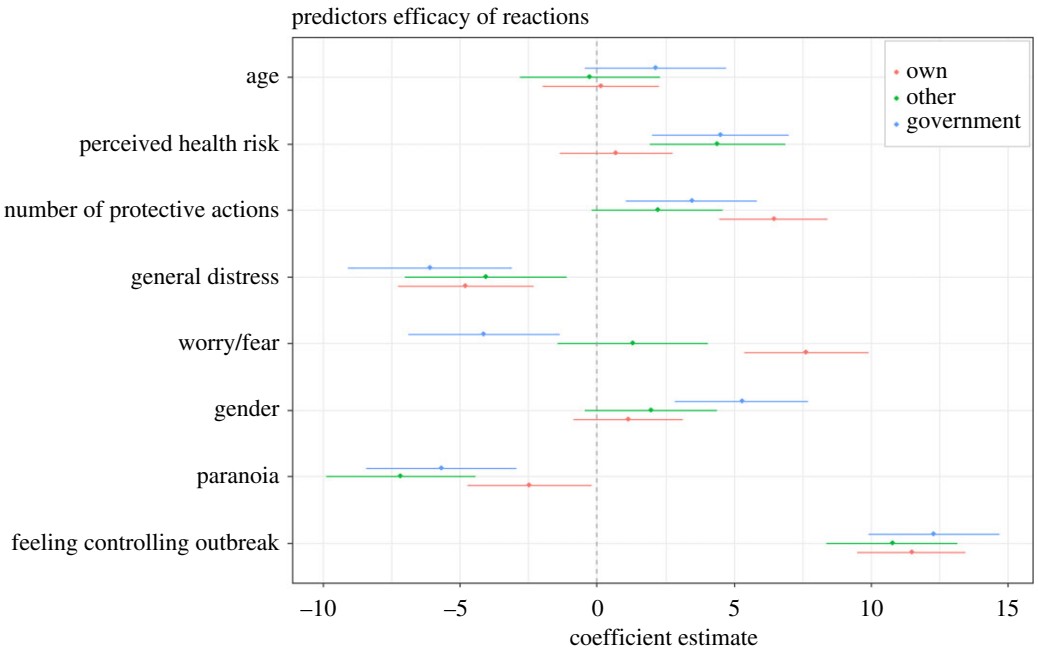

**Figure 6.** Coefficient estimates of the eight predictors for perceived efficacy of self, others' or governmental reaction, with 95% confidence interval. More worry/fear (fear appraisal) increases the efficacy of own reactions but had an opposite effect for rating the efficacy of governmental reactions.

## 3.4. Hypothesis 3: which factors affect the participants' perceived efficacy of the restrictions?

Hypothesis 3 was confirmed. Country significantly affected the perceived efficacy of restrictions, $F_{5,1696} = 77.151$, $p < 0.001$, $\eta^2 = 0.182$ (figure 7). Of the covariates, number of actions related to the efficacy of the

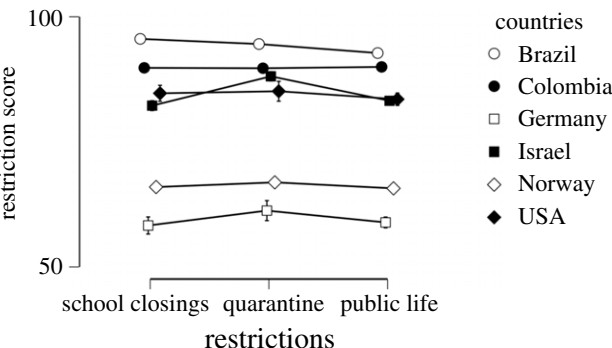

**Figure 7.** Perceived efficacy of restrictions by country. Countries with a severe lockdown also perceived the restrictions as efficient. Norway and Germany, with partial lockdowns, rated them as less efficient.

restrictions: $F_{1,1696} = 31.946$, $p < 0.001$, $\eta^2 = 0.015$, i.e. the more protective actions one performed the more efficacious the perception of the governmental restrictions was. Perceived risk was significant but with a small effect size: $F_{1,1696} = 5.260$, $p = 0.016$, $\eta^2 = 0.002$, i.e. the higher the perceived risk of contagion the higher perceived efficacy. Gender and age had no significant contribution, $p > 0.13$. There was also no difference in the perceived efficacy of the restrictions school closings, quarantine or public life; $F < 1$. All interactions had negligible effect sizes, although restriction by country was significant at $p < 0.001$ and $F_{8.83,2995.168} = 3.221$, $\eta^2 = 0.001$.

## 3.5. Hypothesis 4: which factors contribute to general distress among participants?

Hypothesis 4 was partially confirmed. Seven predictors explained 37.4% of the variance of measured general distress, $F_{8,1630} = 121.283$, $p < 0.001$. We observed an increase in general distress scores the more a respondent also was worried or scared about COVID-19 and the higher their paranoia score (CAPE-P) was. General distress decreased with age (table 7). Neither number of protective actions performed nor perceived risk contributed significantly to distress. There was no effect for gender either. We did not observe any collinearity (VIF below 1.22). We also used the stringency index but this improved the model by only 0.1%, omitted for clarity.

In two explorative follow-up analyses: we (i) included the impact of the six restrictions as predictors (hierarchical regression), and (ii) also investigated whether those who stated being currently affected by the restrictions felt more distressed, regressing out for age, gender, knowledge, feeling of controlling the outbreak, perceived risk, worry/fear, number of actions performed and paranoia.

Including the impact of the six restrictions in the regression model increased the explained variance to 38.2%, a statistically significant but very small improvement. An impact of quarantine on daily life was significant with $\beta = 0.08$, $p < 0.001$, which is a factor 2–4 smaller than age, paranoia and worry/fear had on general distress.

Those who stated being currently affected (85% of our sample) reported higher levels of general distress, $F_{1,1633} = 7.548$, $p = 0.006$, $\eta^2 = 0.004$ compared to those who stated not currently being affected by the outbreak. However, age ($F = 76.6$, $p < 0.001$, $\eta^2 = 0.037$), perceived risk ($F = 21.99$, $p < 0.001$, $\eta^2 = 0.011$), gender ($F = 21.0$, $p < 0.001$, $\eta^2 = 0.01$) and particularly paranoia ($F = 287.56$, $p < 0.001$, $\eta^2 = 0.14$) were larger predictors of general distress than being affected at the time the survey was taken.

# 4. Discussion

The COVID-19 pandemic pushed governments to enact various restrictions to control and constrain the spread of the coronavirus. We reasoned that the perceived efficacy of these measures, as well as the fear of the virus, should serve a key role in the extent to which individuals adhere to these measures [23]. Accordingly, in the current observational study, we measured the overall perceived efficacy of reactions of the self, others and the government, as well as the perceived efficacy of specific restrictions. We also measured the perceived impact of these measures on daily life, the protective actions performed, perceived risk of the virus, worry/fear of contagion, feeling of controlling the outbreak, general distress (opposite of well-being), knowledge about the virus and paranoia. We recruited over 2000 respondents, most residing in six countries.

**Table 7.** Coefficients contributing to general distress (CORE-9 score).

| | unstandardized | s.e. | standardized | t | p | 95% CI lower | 95% CI upper |
|---|---|---|---|---|---|---|---|
| (intercept) | −0.08 | 0.12 | | −0.66 | 0.510 | −0.310 | 0.154 |
| knowledge | 0 | 0 | −0.02 | −0.97 | 0.334 | −0.002 | 0.0006 |
| feeling of controlling outbreak | 0 | 0 | −0.04 | −2.39 | 0.017 | −0.002 | −0.0002 |
| perceived risk | 0 | 7 | 0.00 | 0.44 | 0.659 | -0.001 | 0.002 |
| number of actions | −0.02 | 0.02 | −0.02 | −1.17 | 0.243 | −0.056 | 0.014 |
| worry/fear | 0.01 | 0 | 0.34 | 15.96 | <0.001 | 0.008 | 0.010 |
| gender | 0.06 | 0.03 | 0.03 | 1.87 | 0.061 | −0.003 | 0.124 |
| paranoia | 0.76 | 0.05 | 0.35 | 16.19 | <0.001 | 0.667 | 0.851 |
| age | −0.01 | 0 | −0.18 | −8.52 | <0.001 | −0.013 | −0.008 |

## 4.1. Descriptive findings and perceived severity

The majority of our respondents were directly affected by COVID-19 restrictions like school closings. Respondents rated the concretely listed restrictions as efficient but, at the same time, rated governmental reactions in general as inefficient (figure 2). Indeed, our respondents engaged in protective actions, e.g. hand washing and social distancing. With nearly 90% of our sample engaging in, e.g. social distancing, it is very probable that respondents in our sample supported the need for restrictions and performed them. Although we did not directly ask whether our respondents accepted those restrictions, the self-reported engagement in protective actions and perceiving the restrictions as efficient suggest compliance with these restrictions.

At the country level, Brazil and Colombia experienced elevated levels of distress; people from Colombia (10 years younger, on average, in our sample) also reported high paranoia scores and the lowest score on distinguishing factual knowledge from conspiracy theories about the virus. Since paranoia is higher in younger people [20], we cannot draw conclusions on whether these differences result from the age difference or other factors [55,56].

School and university closings, followed by quarantine and social distancing, were rated as having (or potentially having) the highest impact on people's daily lives. Those who were already experiencing those restrictions rated them as affecting their daily lives more severely, particularly school closings, than people who had not experienced the restrictions yet. The disparity between predictions and actual affect is consistent with canonical findings demonstrating that people provide generally poor predictions of their prospective states [15,16]. As schools can serve as potential epicentres for spreading diseases [57], the disparity between respondents' anticipated impact and actual impact of school closings on daily life can serve as a signal that future efforts of public persuasion campaigns should focus on explaining the necessity of closing schools and on finding alternatives that would reduce the impact on daily lives. Notably, the sudden nature of these changes, as reflected in our sample, which was collected soon after schools were closed in some countries, left no time for respondents to adapt to changes, potentially increasing the disparity between expected and actual experiences. Future investigations should explore how and which individuals adapt to these changes.

## 4.2. Perceived efficacy of restrictions and reactions

Overall, respondents rated their own, others' and their governments' reactions as effective in controlling the COVID-19 outbreak, with their own reactions rated as more effective than reactions of other individuals or their governments. This agrees well with a previous survey finding high belief in the power of guided reactions to prevent influenza [58]. Notably, countries in which participants were dissatisfied with how their government was dealing with the outbreak, e.g. Brazil, Colombia and the USA, rated the efficacy of governmental reactions as lower than that of individuals from the other three countries in our study. The less satisfied one is with one's government the higher was the

reported perceived risk and worry and fear about the Coronavirus [59]. These results are in line with previous studies showing that the efficacy of governmental reactions directly relate to psychological well-being [3]. Furthermore, these perceptions may affect adherence to imposed restrictions, and following advice from health authorities [60] as well as perceiving own reactions as efficient supports preventive health behaviour [58,61].

Perceived efficacy of actions was predicted by feeling of controlling the outbreak as well as general distress. Thus, the more respondents felt that the outbreak could be controlled and the less distressed they were, respectively, the more they perceived their own, others' or governmental reactions as efficacious. Feeling of controlling the outbreak had the largest effect, whereas, somewhat surprisingly, worry and fear about COVID-19 was not a significant predictor. Noteworthy, neither knowledge nor age were significant predictors, which is in line with a recent study showing that experiential and social–cultural factors but not knowledge about the virus and socio-demographic factors contribute to perceived risk [45].

This suggests that personal health concerns about COVID-19 contagion were less important at this early stage of the outbreak for perceived efficacy of reactions, but confidence in a society's ability to fight the outbreak and a general low distress was.

This is further supported by the findings that the number of protective actions performed and perceived risk of contagion also predicted perceived efficacy of reactions [62]; for similar findings, see [48]. Gender also predicted perceived efficacy, particularly efficacy of governmental actions, with women experiencing the reactions as more effective. This is in line with previous research showing higher compliance with protective behaviour among women [49,61]. These findings join a growing number of studies demonstrating that, although COVID-19 might be more lethal for men [63], men tend to devalue health threats compared to women [49,64].

Specific restrictions such as closing schools, quarantine and restrictions on public life were all rated as very effective. We did note a large effect of the country of residency on perceived efficacy of specific restrictions. Countries with less stringent governmental actions at the time of the survey, namely Germany and Norway, had the lowest rating of the efficacy of restrictions. One possible interpretation to this finding might suggest that movement restrictions were perceived as highly effective only after these restrictions were introduced. However, our data are observational and we cannot draw any causal conclusion, so interpretations should be made with caution. Nonetheless, our results pertaining to the effect on people's lives, i.e. the perceived severity of restrictions, seem to corroborate this line of reasoning.

## 4.3. Acceptance versus perceived efficacy of restrictions and reactions

Considering the perceptions of various restrictions, one can draw a theoretical dissociation between accepting a social restriction and its perceived effectivity (we thank an anonymous reviewer for highlighting this distinction). For instance, people might see closing restaurants as an acceptable measure but simultaneously they might consider it ineffective to control the spread of the outbreak. Similar broader theoretical distinctions have been previously proposed, for instance in a classical paper [65] between following prescriptions versus agreeing to the outcomes of social norms.

Although we did not directly measure the acceptability of the restrictions, our data hints to this theoretical distinction (i.e. table 6 and figure 4). Our exploratory analysis presented in table 6 also showed that for the whole sample perceived risk and worry/fear better predicted perceptions that their country was not doing enough to handle the outbreak. In fact, previous research has shown that people have biased perceptions of health risks [66]. For instance, when they misperceive exponential coronavirus spread as a linear trend [18]. Accordingly, when people appropriately understand exponential coronavirus spread, they experience more fear [30].

Future studies could take advantage of our findings to disentangle acceptance of social prescriptions from their perceived effectivity by further using questions in which respondents rate the perceived efficacy and acceptability of their own, others' and governmental reactions. In such cases, other motivational incentives, not necessarily related to negative emotions, such as monetary rewards or social status, can also work well within a similar trade-off model. In laboratory settings, coordination games (e.g. [67]) have used analogous strategies to study how social norms influence behaviour.

## 4.4. Effect of COVID-19 on psychological distress

Across our sample, general distress was higher the more paranoid, worried and scared a respondent was [22]. Similarly, reduced beliefs about the controllability of the outbreak and being younger also predicted

higher distress scores. Perceived risk of contagion was not a significant predictor of distress. This might indicate that participants were distressed about concerns partially unrelated to the health-related impacts of COVID-19. Indeed, economic fear and worry about relatives were mentioned as prominent impacts by many respondents. Importantly, in mid-March the reported number of COVID-19 cases and deaths was relatively small, but rose considerably as the days went by. This exponential increase in cases and deaths, if understood [18], may also have contributed to uncertainty, feeling worried about getting infected with COVID-19 and fear of contagion.

General distress in some countries was higher than what is common in the general population. Respondents from Brazil, Colombia and the USA had elevated levels of distress, whereas respondents from Israel, Norway and Germany had comparatively lower levels of distress. Notably, the last three countries were also more satisfied with how their country fought the outbreak.

## 4.5. Cross-country heterogeneity

Our study was not set up focusing on differences between the six countries. However, it is undeniable that the six countries differ on key aspects, e.g. general trust in government [68], which has been shown to affect compliance with public health advice (e.g. during the Ebola outbreak in Liberia, [46], and risk perception [37,45]; Norway was not included in this study). In this section, we speculate about these differences.

Risk perceptions have been previously found to correlate with adherence to preventive health behaviours in a cross-national survey on the COVID-19 pandemic [45,69], which is also supported by our findings that risk perceptions contribute to the perceived efficacy of actions. However, we found that the latter was either predicted by country or risk perceptions. This may be due to the fact that the countries included in our sample seem to differ in risk perceptions, and perhaps the individual measures reflect this tendency.

Country affected how people perceived the efficacy of restrictions issued by their governments. Interestingly, this did not necessarily reflect the governments' actual efficacy, as the countries whose inhabitants rated their government reactions as less effective were Germany and Norway, two countries that had followed the WHO guidelines from the start of the outbreak. Further, in those two countries as well as Israel, our respondents did state that their country did enough to fight the outbreak (figure 5).

Across analyses, feeling of controlling the outbreak and the number of protective actions one performed predicted the perceived efficacy of actions and perceived efficacy of restrictions, respectively. One way of interpreting these findings is by looking into the locus of control theory. People with high locus or sense of control tend to engage in health-promoting behaviour [70]. However, we measured feeling of control with one sentence referring to the collective (I feel we can control the outbreak), which includes measures outside of the control of the individual. There is evidence that more individualistic countries tend to attribute more negative connotations to external locus of control than more collectivistic ones [71]. In our findings, Norway, Germany and the USA (more individualistic) had relatively lower scores of feeling of controlling the outbreak than Colombia, Brazil and Israel (more collectivistic countries; Israel being a mix of both dimensions) [72].

These are only speculations, and one must be careful drawing inferences from cross-national samples. We do not claim, for example, that our samples are representative of these countries. Future studies could control for cultural differences in e.g. trust in government and risk perception in countries with similar strategies to deal with this pandemic or similar global crises.

## 4.6. Limitations

Several limitations potentially compromise the generalizability of our results. We used a snowball sampling method that did not succeed in recruiting a sufficient number of respondents from some target populations (Italian and Arabic-speaking respondents). Furthermore, the survey did not target a representative sample, and we did not probe socio-economic status to control for potential interactions. Moreover, some of our samples (e.g. respondents from Colombia) were predominantly students. However, the pandemic affects all members of the community, and we believe that response differences by socio-economic status and education might be seen at a later stage of the pandemic but not necessarily at this early stage. In line with this argumentation, a recent study found that socio-cultural but not socio-demographic factors influence risk perception [45].

It is possible that there are cultural differences in translations of the scales and general mental health across countries, which we could not control for. It is also possible that we had a sampling bias, due to

people who were feeling distressed wanting to participate in a survey about COVID-19 and how people were affected. Indeed, our results did seem to indicate higher general distress in some countries than what is usually reported in the general population. This is also reflected in a higher proportion of participants being dissatisfied with how their country fights the outbreak. Few people are unaffected by the pandemic, and hence increased distress levels were not surprising given the severity of the situation, drastic changes in people's lives and governmental restrictions.

In addition to existing scales, we created new items to measure several constructs, e.g. perceived efficacy of restrictions and reactions. Regarding feeling of control we formulated it as 'I feel we can control the outbreak' whereas efficacy of own action was formulated as 'my actions are effective in limiting the outbreak'. As our results show, these two items did not correlate highly, and 'we can control' was rated lower than the efficacy of own reactions, suggesting that non-personal efficacy was judged as lower than self-efficacy [23]. However, minor differences in formulations or items can lead to different outcomes for the same research questions [73].

Our survey was designed to measure responses at an early phase of the outbreak. Indeed, 13–59% of respondents in Europe, Israel and the USA indicated that they were not (yet) affected by the outbreak. By contrast, at the launch of the survey in Colombia the entire city of Bogota was already in lockdown. The different phases of the outbreak had drastic effects on measured well-being, as participants from Colombia and Brazil reported more distress and paranoia than participants from Germany and Norway.

We did not directly measure trust in one's government, although trust is a key determinant of resilience in pandemics and fosters compliance with the restrictions [24]. However, we did ask how much our respondents thought their government is doing enough. Mistrust could be reflected in answering no.

Furthermore, we did not explicitly assess whether respondents believed the restrictions and reactions to the pandemic are acceptable. Thus, respondents could rate some restrictions low on efficacy either because they believed these restrictions are not efficient, or because such restrictions are not acceptable. Our exploratory analyses (table 6 and figure 5) hints that these two theoretical distinctions are related. Specifically, we show that the more efficacious they perceived their own reaction, the more they thought their country is doing enough (i.e. that their country's reactions are acceptable). Furthermore, the less they perceived governmental reactions as efficacious, the less they thought their country is doing enough. Future studies should disentangle these constructs and test, for example, situations in which action is needed but proposed restrictions are deemed irrelevant.

Finally, the current data present a multiverse of analysis options, as some researchers might see paranoia or feeling of control as outcome variables and not predictors. We provide anonymized data for other researchers to peruse [74,75] and future meta-analyses and cross-national analyses to learn and be better prepared for similar future events [4,40,74].

## 4.7. Implications

The novel SARS-CoV-2 virus required governments to take urgent measures to mitigate the epidemic, since no pharmaceutical treatments and vaccines exist yet. SARS-CoV-2 is also hard to control with intensive testing, isolation and tracing [76]. Therefore, governments applied physical distancing to limit the transmission rates of the virus [77]. Our findings indicate that citizens cope better with governmental-issued restrictions if citizens believe they are effective in limiting the outbreak. Furthermore, our findings suggest that inducing a sense of control might assist in boosting the perceived efficacy of centrally issued restrictions. Relatedly, in New Zealand, a country acclaimed for the way it dealt with the pandemic [40], researchers found that trust in science, potentially related to sense of control, predicted compliance with restrictions [68]. As the COVID-19 pandemic is currently predicted to linger for an extended period of time, possibly even a few years [76], it is pertinent for governments and health organizations to facilitate public endorsement of restrictions aimed to limit the outbreak. Notably, our findings transcend the specifics of individual restrictions and countries. Our results suggest that transparent and persuasive communication about the efficiency of measures might be the key to gain adherence and by that reduce the detrimental effects on mental health.

## 5. Summary and conclusion

In an international survey conducted at an early phase of the COVID-19 pandemic, we found that people in various countries perceived restrictions as being effective, and they rated their own reactions as

effective in limiting the outbreak. Overall, respondents perceived reactions and restrictions as more efficacious when they reported higher feelings of 'we can control the outbreak' and higher perceived risk of contracting the virus. Dissatisfaction with the reactions of respective governments was associated with heightened perception of risk and increased worry and fear, as well as paranoia. Increased worry and fear, in turn, were associated with heightened distress, and a significant portion of our respondents were experiencing clinical levels of general distress already in the very early days and weeks of the lockdown. Lower ratings of efficacy of governmental reactions were associated with higher paranoia scores, a worrying finding given the rise of conspiracy theories and social destabilizations. Furthermore, our results suggest that behaviour, gauged in our survey with items asking about performed actions, was driven by perceived risk of contagion, whereas well-being and distress were driven by worry and fear about contagion.

Together, these findings highlight several factors that play a key role in psychological reactions to the outcomes of the early phases of COVID-19 pandemic. In the face of drastic changes to daily habits, beliefs in efficacious governmental and personal responses and the controllability of the pandemic emerge as factors protecting the well-being of citizens across the world. As these factors concomitantly contribute to adherence to institutional restrictions [4], our results highlight the importance of efforts to convincingly communicate the efficacy of restrictions to the general public. Successfully controlling the outbreak requires understanding the psychology behind adherence with the restrictions and compliance with public health advice, respectively.

Ethics. The study was approved by the local ethics board at UiT (2017/2019), by the Norwegian Centre for research data (ref. no. 287376). The study was also approved by the local ethics committee at Ben Gurion University and by the ethics committee from 'Facultad de Ciencias Humanas' Universidad Nacional de Colombia, Sede Bogotá (Ethical approval no. B.VIE-FCH-21–2020). Data collection in Brazil followed the recommended local ethics guidelines for online survey studies in social sciences and humanities [78].

Data accessibility. The materials, data and the pre-registered analysis plan can be found at https://osf.io/bh2cz/ and doi:10.17605/OSF.IO/BH2CZ.

Authors' contributions. M.J.M., K.K. and G.P. conceived, designed and coordinated the study. G.P. analysed the data. All authors collected data, helped interpret the data, drafted the manuscript and gave their final approval for publication.

Competing interests. We declare we have no competing interests.

Funding. We received no funding for this study.

Acknowledgements. The authors would like to thank Aviv Mokady and Ofir Lugassy for their help in translating survey materials to Hebrew and contributing to data collections, and to Sewar Abu Freih for her help in translating to Arabic. We also thank Giovanni Briganti for translating the survey to Italian.

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
