## [Reviewer comments · Royal Society Open Science]

Review History

RSOS-200644.R0 (Original submission)

Review form: Reviewer 1

Is the manuscript scientifically sound in its present form?

No

Are the interpretations and conclusions justified by the results?

Yes

Is the language acceptable?

No

Do you have any ethical concerns with this paper?

No

Have you any concerns about statistical analyses in this paper?

Yes

Recommendation?

Accept with minor revision (please list in comments)

Comments to the Author(s)

Overall, the novelty, sample, and scope of the manuscript is quite promising, given its data ranging over six countries, tapping onto the perceptions of people on a larger scale, developing multilingual measures, and, a cross cultural study which facilitates better view of mental health impact of citizens globally in such a small span of time.

However, the weakest part of of the paper is its writing style right from the introduction. There are no sub-headings, either in introduction or in the result, leading to more cognitive load on the reader, followed by a weak discussion.

The introduction lacks compelling insights-- a brief status report of the COVID-19 in all the six countries should be mentioned one by one with relevant data, so that a reader can have a brief idea of all the six countries. The title of the manuscript state two psychological variables, but, in the variables section, plethora of variables appear without any proper operationalisation, rationalisation, and presentation. It should be properly mentioned as to how and why in the introduction sections, the authors have selected so and so variables. In short, the existing framework fell short in explaining study variables properly and needs refinement. Variables need to be properly defined operationally. Besides, only one question measure feeling of control. What is operationalisation of the construct 'control' in the study?

The result section is appear vague and with certain wrong interpretations. For ex - the last line of Page 9, it is mentioned that feeling of control and perceived risk was similar Germany and US, but the data in the table reflects differently. Similarly, on page 10, worry and perceived risk was higher than perceived risk, except Norway, which is not in sync with the data. In spite of the data being rich, putting them in a single-go, make it hard for the reader to comprehend the meaning easily. The way hypothesis have been derived is not properly written.

The discussion is too generic and lacks compelling arguments, reasoning, and theoretical underpinnings. For ex - Even if the gender and age has been considered, the reason for their findings were not provided. Given the study encompasses six countries, country-specific cultures can also play a huge role for the findings and need explicit mentioning in the discussion.

In spite of the manuscript bearing a huge sample and a much needed cross-cultural comparison of COVID-19, a better version will further enrich the scope and citation of the paper in future.

Review form: Reviewer 2**Is the manuscript scientifically sound in its present form?**

No

Are the interpretations and conclusions justified by the results?

No

Is the language acceptable?

Yes

Do you have any ethical concerns with this paper?

No

Have you any concerns about statistical analyses in this paper?

No

Recommendation?

Major revision is needed (please make suggestions in comments)

Comments to the Author(s)

I had mixed impressions of the manuscript. I think that the data collected by the authors, given its multi-national nature could provide important insights about perceptions of the pandemic (and their country's responses to it), which could be informative to policy makers and researchers examining the impact of COVID-19 alike. Indeed, I think multi-national endeavors to document the psychological impact of COVID-19 across different nations are critical. For instance, the descriptive statistics provided in this paper are very informative and useful. I also commend the authors for their transparency in describing their work (i.e., stating which analyses were exploratory) and for pre-registering the research on the OSF. However, at the same time, I also have some serious concerns about the manuscript and the analyses as they are currently presented. I found myself overwhelmed reading the paper, given the heterogeneity of different measures considered, and the hypotheses being tested. The theoretical rationale for the different hypotheses, and justifications for which variables were included, was not clear to me. Because of this, I think the important contributions which this paper could potentially make are presently being undermined.

Below I try to highlight some of my big picture concerns on this matter, and then also, I list some more detailed comments/questions I had as I read through the paper. Again though, I think the data presented in this paper has potential, and with careful reflection, re-writing, and revision, could make important theoretical and practical contributions to our understanding of the psychological impact of COVID-19.

I wish the authors all the best with this important work going forward.

Major Comments

Theoretically justifying predictions:

It was not clear to me why some of the variables were being included either as predictors or DVs. For example, in Table 3 and Table 4 the authors regress several factors onto perceived efficacy of actions. I believe that in both Table 3 and 4 this was the combined efficacy score of personal actions, country actions, and national actions although I was not sure because in Table 3 title is just said "efficacy of actions" while in Table 4 it explicitly said personal/country/other. I can understand why personal control might relate to perceived efficacy of individual actions – although the authors need to review more literature on why this would be true – but I am less clear on why personal control would predict perceived efficacy of the nation/or others' actions. Even if I have a high sense of personal control, I might think that my country is not effective? Other variables which were included as predictors were also less clear to me. For example, why would my level of paranoia relate to my perceived efficacy of actions like social distancing? For every predictor that is included in the regression models, it will help for the authors to justify on the basis of past literature why it would / or would not be expected to relate to the DV being predicted.

I had similar questions about the justification of predictors included to predict psychological distress. For one, I wondered if worry/fear is itself an index of distress? Can the authors provide justification for why these two variables should be treated separately? Similarly, the clinical measure of paranoia seemed somewhat related to distress as well, or at least, should be treated more as a DV? If we look at the effect sizes of these two predictors of general distress they are also markedly higher than the other predictors, which added to my concern about their distinctions from the DV. Moreover, in the introduction, p.3, lines -31-34 the authors note briefly

“anxiety and paranoia have been shown to develop under conditions of worry and perceived loss of control”. Here, paranoia seems to be presented as a DV developing during conditions of uncertainty. Also, it seems a little circular to me to state that anxiety develops from worry because worry is a core component of anxiety? Paranoia was also described as a DV on p. 56 where the authors write: “As the COVID-19 outbreak poses a severe threat which can cause physical, psychological and social harm, it can enhance long-term anxiety, which, in turn, can lead to maladaptive outcomes such as paranoia”.

I also wondered how perceived knowledge about the pandemic and belief in conspiracy theories fit into the authors overarching theoretical framework. Can they provide justification for why it makes sense to compute a difference score of knowledge about COVID-19 from perceived conspiracies? Does past literature do this? I imagine that I might know very little about a subject, but also, be very skeptical of conspiracy theories. I would want to also see a review/discussion of past literature describing the link between knowledge/conspiracy theories and perceived efficacy of self/other/government actions and distress. Why should we expect knowledge/conspiracy theories to relate to these outcomes? I am guessing that the authors conceptualize this measure as an index of uncertainty (a construct they discuss on p.3 lines 31-36) but this could be made clearer.

Refining the Introduction:

I think a lot of the concerns I raised above can be addressed by re-working the introduction and the theoretical framework presented there. First, I think there is some literature reviewed here which does not really relate to what was measured in the actual research. Most notably is the discussion on p.3 to p.4 about uncertainty leading to pro-sociality - I was confused about this, because I don't think the authors measure pro-sociality.

In the introduction, it would be helpful for the authors to clearly operationalize and define all the IVs and DVs they will focus on, and explain why they would be expected to relate. For example, the authors talk a lot about “control” but they never really define it. Psychological control has a broad literature associated with it, and has been defined in different ways (e.g., thinking I can influence or control others, or resist influence from others; internal/external locus of control ect). Perceptions of control are also sometimes operationalized as part of efficacy/competence. So, it will be helpful to know exactly what the authors mean by control, and how in their framework it is distinct from efficacy. Also, the authors frequently discussed uncertainty stemming from a pandemic, but they never explicitly say they measured uncertainty in their research. Thus, I want to know how uncertainty might have been operationalized in their work (if it was): perhaps it was the knowledge measure? , or the paranoia measure?, or the worry/fear measure? All of this needs to be made more clear.

Lastly, I think it could be critical for the authors to flesh out the four predictions they quickly summarize on p.4 (lines 54-59) and then re-state as hypotheses on p.6-7. First, I suggest that these two sections could be merged into one section and just appear at the end of the introduction. Second, there is never any justification of “why” provided for any of these predictions. For example, why would we expect that people who are actually experiencing countermeasures will perceive them as more impactful, then how people who have not yet experienced them might imagine them to be?

Other comments:

- The authors often use the term “mediate” but I am not sure this term is being used correctly in the paper. They seem to be using the term to mean “predict”. For example, on p. 13 they write “ which factors mediate the participants’ perceived efficacy of the countermeasures?”. But mediation refers to one variable “M” explaining/underlying the relation between to other variables X and Y. Its not clear to me what X,M,Y are in this case.

- I was confused by the authors' decision to measure both perceived efficacy of government actions and also countermeasures initiated by the government. Are these not the same thing? The authors themselves even say something to this extent on p. 7, line 58 " We did not include the efficacy of countermeasures in this analysis to avoid redundancy, as the perceived efficacy of actions score includes actions by government". Why measure two very similar things, and use one versus the other across different analyses?
- As I noted above I think the decision to combine personal, other, government efficacy into one measure to test hypothesis 2 needs more justification. I could see these being quite distinct, and indeed, these are what the descriptive results presented by the authors shows.
- I would like to see more consistency in how the variables are referred to across the different tables. For example, sometimes the authors wrote CORE-9/distress (Table 2) and other times just CORE-9 (Table 3. Also, why is CAPE-P not referred to as CAPE-P/paranoia in Table 2, similar to CORE-9/distress? Personally, I think it is easier to follow when variables are referred to by the construct name (e.g., distress, paranoia) versus the scale name (CORE-9, CAPE-P).
- In the results section, I appreciated how the authors reminded us what each hypothesis was : e.g., " Hypothesis 4: Which factors contribute to general distress among participants?" but I think it may be easier to follow if they frame this as an actual hypothesis, (A,B,C will predict general distress) versus a question.
- On p. 9, I was confused by this statement: "On the other hand, 235 (10.4%) indicated that they did not perform any protective actions, and 201 (9.2%) answered that they did not perform any action to protect others. There were 111 participants doing neither." How could there be 235 people who did not perform any protective actions, but then also, 111 participants doing neither?
- I found the variable name "effect of countermeasures" confusing – this sounds a lot like "impact" of countermeasures. Maybe relabel to "Types of countermeasures initiated by government". It would also be helpful to explicitly say that participants could select more than one option here.
- I am not sure that the Tables are using APA style

Decision letter (RSOS-200644.R0)

Dear Dr Pfuhl,

The editors assigned to your paper ("Perceived efficacy of actions and feelings of distress during the early phase of the COVID-19 outbreak in six countries") have now received comments from reviewers. We would like you to revise your paper in accordance with the referee and Associate Editor suggestions which can be found below (not including confidential reports to the Editor). Please note this decision does not guarantee eventual acceptance.

Please submit a copy of your revised paper before 12-Jun-2020. Please note that the revision deadline will expire at 00.00am on this date. If we do not hear from you within this time then it will be assumed that the paper has been withdrawn. In exceptional circumstances, extensions may be possible if agreed with the Editorial Office in advance. We do not allow multiple rounds of revision so we urge you to make every effort to fully address all of the comments at this stage.

If deemed necessary by the Editors, your manuscript will be sent back to one or more of the original reviewers for assessment. If the original reviewers are not available, we may invite new reviewers.

- Data accessibility

If you wish to submit your supporting data or code to Dryad (<http://datadryad.org/>), or modify your current submission to dryad, please use the following link:
<http://datadryad.org/submit?journalID=RSOS&manu=RSOS-200644>

- Competing interests

- Authors' contributions

- Acknowledgements

- Funding statement

on behalf of Dr Christina Demski (Associate Editor)
openscience@royalsociety.org

Associate Editor's comments (Dr Christina Demski):

Comments to the Author:

Both reviewers see merit in the dataset and manuscript but would like to see revisions to the introduction and discussion (predominantly). It would therefore be good to see some more specific discussion of the theoretical framework underlying the analysis. Please address all comments by the reviewers.

Reviewers' Comments to Author:

Reviewer: 1

Comments to the Author(s)

Overall, the novelty, sample, and scope of the manuscript is quite promising, given its data ranging over six countries, tapping onto the perceptions of people on a larger scale, developing multilingual measures, and, a cross cultural study which facilitates better view of mental health impact of citizens globally in such a small span of time.

However, the weakest part of of the paper is its writing style right from the introduction. There are no sub-headings, either in introduction or in the result, leading to more cognitive load on the reader, followed by a weak discussion.

The introduction lacks compelling insights-- a brief status report of the COVID-19 in all the six countries should be mentioned one by one with relevant data, so that a reader can have a brief idea of all the six countries. The title of the manuscript state two psychological variables, but, in the variables section, plethora of variables appear without any proper operationalisation, rationalisation, and presentation. It should be properly mentioned as to how and why in the introduction sections, the authors have selected so and so variables. In short, the existing

framework fell short in explaining study variables properly and needs refinement. Variables need to be properly defined operationally. Besides, only one question measure feeling of control. What is operationalisation of the construct 'control' in the study?

The result section is appear vague and with certain wrong interpretations. For ex - the last line of Page 9, it is mentioned that feeling of control and perceived risk was similar Germany and US, but the data in the table reflects differently. Similarly, on page 10, worry and perceived risk was higher than perceived risk, except Norway, which is not in sync with the data. In spite of the data being rich, putting them in a single-go, make it hard for the reader to comprehend the meaning easily. The way hypothesis have been derived is not properly written.

The discussion is too generic and lacks compelling arguments, reasoning, and theoretical underpinnings. For ex - Even if the gender and age has been considered, the reason for their findings were not provided. Given the study encompasses six countries, country-specific cultures can also play a huge role for the findings and need explicit mentioning in the discussion.

In spite of the manuscript bearing a huge sample and a much needed cross-cultural comparison of COVID-19, a better version will further enrich the scope and citation of the paper in future.

Reviewer: 2

Comments to the Author(s)

I had mixed impressions of the manuscript. I think that the data collected by the authors, given its multi-national nature could provide important insights about perceptions of the pandemic (and their country's responses to it), which could be informative to policy makers and researchers examining the impact of COVID-19 alike. Indeed, I think multi-national endeavors to document the psychological impact of COVID-19 across different nations are critical. For instance, the descriptive statistics provided in this paper are very informative and useful. I also commend the authors for their transparency in describing their work (i.e., stating which analyses were exploratory) and for pre-registering the research on the OSF. However, at the same time, I also have some serious concerns about the manuscript and the analyses as they are currently presented. I found myself overwhelmed reading the paper, given the heterogeneity of different measures considered, and the hypotheses being tested. The theoretical rationale for the different hypotheses, and justifications for which variables were included, was not clear to me. Because of this, I think the important contributions which this paper could potentially make are presently being undermined.

Below I try to highlight some of my big picture concerns on this matter, and then also, I list some more detailed comments/questions I had as I read through the paper. Again though, I think the data presented in this paper has potential, and with careful reflection, re-writing, and revision, could make important theoretical and practical contributions to our understanding of the psychological impact of COVID-19.

I wish the authors all the best with this important work going forward.

Major Comments

Theoretically justifying predictions:

It was not clear to me why some of the variables were being included either as predictors or DVs. For example, in Table 3 and Table 4 the authors regress several factors onto perceived efficacy of actions. I believe that in both Table 3 and 4 this was the combined efficacy score of personal actions, country actions, and national actions although I was not sure because in Table 3 title is just said "efficacy of actions" while in Table 4 it explicitly said personal/country/other. I can understand why personal control might relate to perceived efficacy of individual actions -

although the authors need to review more literature on why this would be true – but I am less clear on why personal control would predict perceived efficacy of the nation/or others' actions. Even if I have a high sense of personal control, I might think that my country is not effective? Other variables which were included as predictors were also less clear to me. For example, why would my level of paranoia relate to my perceived efficacy of actions like social distancing? For every predictor that is included in the regression models, it will help for the authors to justify on the basis of past literature why it would / or would not be expected to relate to the DV being predicted.

I had similar questions about the justification of predictors included to predict psychological distress. For one, I wondered if worry/fear is itself an index of distress? Can the authors provide justification for why these two variables should be treated separately? Similarly, the clinical measure of paranoia seemed somewhat related to distress as well, or at least, should be treated more as a DV? If we look at the effect sizes of these two predictors of general distress they are also markedly higher than the other predictors, which added to my concern about their distinctions from the DV. Moreover, in the introduction, p.3, lines -31-34 the authors note briefly "anxiety and paranoia have been shown to develop under conditions of worry and perceived loss of control". Here, paranoia seems to be presented as a DV developing during conditions of uncertainty. Also, it seems a little circular to me to state that anxiety develops from worry because worry is a core component of anxiety? Paranoia was also described as a DV on p. 56 where the authors write: "As the COVID-19 outbreak poses a severe threat which can cause physical, psychological and social harm, it can enhance long-term anxiety, which, in turn, can lead to maladaptive outcomes such as paranoia".

I also wondered how perceived knowledge about the pandemic and belief in conspiracy theories fit into the authors overarching theoretical framework. Can they provide justification for why it makes sense to compute a difference score of knowledge about COVID-19 from perceived conspiracies? Does past literature do this? I imagine that I might know very little about a subject, but also, be very skeptical of conspiracy theories. I would want to also see a review/discussion of past literature describing the link between knowledge/conspiracy theories and perceived efficacy of self/other/government actions and distress. Why should we expect knowledge/conspiracy theories to relate to these outcomes? I am guessing that the authors conceptualize this measure as an index of uncertainty (a construct they discuss on p.3 lines 31-36) but this could be made clearer.

Refining the Introduction:

I think a lot of the concerns I raised above can be addressed by re-working the introduction and the theoretical framework presented there. First, I think there is some literature reviewed here which does not really relate to what was measured in the actual research. Most notably is the discussion on p.3 to p.4 about uncertainty leading to pro-sociality - I was confused about this, because I don't think the authors measure pro-sociality.

In the introduction, it would be helpful for the authors to clearly operationalize and define all the IVs and DVs they will focus on, and explain why they would be expected to relate. For example, the authors talk a lot about "control" but they never really define it. Psychological control has a broad literature associated with it, and has been defined in different ways (e.g., thinking I can influence or control others, or resist influence from others; internal/external locus of control ect). Perceptions of control are also sometimes operationalized as part of efficacy/competence. So, it will be helpful to know exactly what the authors mean by control, and how in their framework it is distinct from efficacy. Also, the authors frequently discussed uncertainty stemming from a pandemic, but they never explicitly say they measured uncertainty in their research. Thus, I want to know how uncertainty might have been operationalized in their work (if it was): perhaps it was the knowledge measure?, or the paranoia measure?, or the worry/fear measure? All of this needs to be made more clear.

Lastly, I think it could be critical for the authors to flesh out the four predictions they quickly summarize on p.4 (lines 54-59) and then re-state as hypotheses on p.6-7. First, I suggest that these two sections could be merged into one section and just appear at the end of the introduction. Second, there is never any justification of “why” provided for any of these predictions. For example, why would we expect that people who are actually experiencing countermeasures will perceive them as more impactful, then how people who have not yet experienced them might imagine them to be?

Other comments:

- The authors often use the term “mediate” but I am not sure this term is being used correctly in the paper. They seem to be using the term to mean “predict”. For example, on p. 13 they write “which factors mediate the participants’ perceived efficacy of the countermeasures?”. But mediation refers to one variable “M” explaining/underlying the relation between to other variables X and Y. Its not clear to me what X,M,Y are in this case.

- I was confused by the authors’ decision to measure both perceived efficacy of government actions and also countermeasures initiated by the government. Are these not the same thing? The authors themselves even say something to this extent on p. 7, line 58 “ We did not include the efficacy of countermeasures in this analysis to avoid redundancy, as the perceived efficacy of actions score includes actions by government”. Why measure two very similar things, and use one versus the other across different analyses?

- As I noted above I think the decision to combine personal, other, government efficacy into one measure to test hypothesis 2 needs more justification. I could see these being quite distinct, and indeed, these are what the descriptive results presented by the authors shows.

- I would like to see more consistency in how the variables are referred to across the different tables. For example, sometimes the authors wrote CORE-9/distress (Table 2) and other times just CORE-9 (Table 3. Also, why is CAPE-P not referred to as CAPE-P/paranoia in Table 2, similar to CORE-9/distress? Personally, I think it is easier to follow when variables are referred to by the construct name (e.g., distress, paranoia) versus the scale name (CORE-9, CAPE-P).

- In the results section, I appreciated how the authors reminded us what each hypothesis was : e.g., “ Hypothesis 4: Which factors contribute to general distress among participants?” but I think it may be easier to follow if they frame this as an actual hypothesis, (A,B,C will predict general distress) versus a question.

- On p. 9, I was confused by this statement: “On the other hand, 235 (10.4%) indicated that they did not perform any protective actions, and 201 (9.2%) answered that they did not perform any action to protect others. There were 111 participants doing neither.” How could there be 235 people who did not perform any protective actions, but then also, 111 participants doing neither?

- I found the variable name “effect of countermeasures” confusing – this sounds a lot like “impact” of countermeasures. Maybe relabel to “Types of countermeasures initiated by government”. It would also be helpful to explicitly say that participants could select more than one option here.

- I am not sure that the Tables are using APA style

Author's Response to Decision Letter for (RSOS-200644.R0)

See Appendix A.

RSOS-200644.R1 (Revision)

Review form: Reviewer 1

Is the manuscript scientifically sound in its present form?

Yes

Are the interpretations and conclusions justified by the results?

Yes

Is the language acceptable?

Yes

Do you have any ethical concerns with this paper?

No

Have you any concerns about statistical analyses in this paper?

No

Recommendation?

Accept with minor revision (please list in comments)

Comments to the Author(s)

The revised manuscript seems to well-cover its objectives and breadth with the minute modifications carried out. The flow of the writing and presentation is much clear and making a coherent sense. However, few rectifications are still needed before giving it a final touch:

1. The formulation of the hypothesis: Few more deliberations needed for strong positioning of all the hypotheses. The author may include further studies.
2. The Discussion: The discussion segment have been enriched. However, the ground reality behind the differences of perception among men and women in the present sample regarding the efficacy and a more gripping paragraph on findings related to the cultural differences among the two set of countries may yield more findings and extend the depth of the manuscript.
3. On the implication part, the leadership of South Korea and New-Zealand may be cited with the fruitful results as they adhere to trust, transparent and persuasive communication. Role of social media may also be highlighted upon in the case of awareness, minus the fake news.

Review form: Reviewer 2

Is the manuscript scientifically sound in its present form?

Yes

Are the interpretations and conclusions justified by the results?

Yes

Is the language acceptable?

Yes

Do you have any ethical concerns with this paper?

No

Have you any concerns about statistical analyses in this paper?

No

Recommendation?

Accept with minor revision (please list in comments)

Comments to the Author(s)

This was my second time reviewing the manuscript "Perceived efficacy of countermeasures and actions and their impact on mental health during the early phase of the COVID-19 outbreak in six countries". I was Reviewer 2 in the initial review.

Overall, I think that the revised version of the manuscript is much improved compared to the initial draft, and I appreciate the efforts which the authors took to incorporate my feedback, as well as that of the other reviews. For example, I think that the introduction and discussion are now better organized and more clearly set up the theoretical hypotheses tested in the paper.

This said, there were still a few aspects of the manuscript I was unsure about (some of which are similar to concerns I had in my original review). I describe these below in order of how they appear in the manuscript, but leave it to the editor and the authors to decide what should be implemented or not.

I wish the authors all the best with this important work.

Intro

1. On p.4, lines 36 to 47, I like how the authors note research describing how people are poor predictors of the future and of how they'll actually experience something. However, here the authors seem to assume that people are biased to be overly optimistic about the future, and I wondered if there is any previous research which might suggest that people could also be potentially overly pessimistic?

2. In the introduction, p.5, lines 14-30, the authors make a case for why perceived risk/threat of the virus would predict perceived efficacy of the restrictions used to reduce virus spread. However, here the authors seem to confound acceptability/necessity of the restriction, with the effectiveness of the intervention. But I am not sure if these are one in the same: I might think there is great need to for an intervention/restriction, but also think that the interventions/restrictions presently offered are not effective/efficacious. Perhaps the authors could note this theoretical distinction in the General Discussion, and say how future work could better disentangle these perceptions.

3. Throughout the manuscript I still got confused between the distinction of perceived efficacy of actions (performed by individuals, others, government) and countermeasures proposed by government. This first becomes problematic when understanding how H2 is distinct from H3 on p.7. Moreover, throughout the manuscript, the authors seem to oscillate between them fairly quickly and interchangeably: e.g., see p. 18, line 20: "The majority of our respondents were directly affected by COVID-19 countermeasures. They engaged in protective actions" Here, the authors seem to refer to countermeasures as protective actions, so I am not sure what is what. Maybe, the authors could use more distinct names, or have a paragraph further explaining the distinction between the two sets of constructs?

Study

4. On p. 8, could the authors clarify that for the impact of countermeasures participants rated impact even if they did not experience the countermeasure (i.e., anticipated impact). I think based on results, that this is what is actually done but this is not stated clearly in the manuscript right now. Also, if participants did not experience something, wouldn't the rating scales have to be framed differently "I would expect this to impact my daily life a lot?"

5. I still am not convinced with how the authors justify using feeling of control operationalized as "I feel we can control the outbreak of the Coronavirus" as a predictor of efficacy of actions operationalized as "My/Our actions are effective in limiting the outbreak". I find that this is the same construct – I would like to know what these items correlate at. Also, assuming they were different, I could see efficacy, thinking actions are effective to limited the diseases spread, predicting (rather than being predicted by) a sense that the country can control the outbreak. Maybe discuss this in the general discussion as a potential limitation.

6. P.12, line 56, I find the phrase: "Were personal life- and work-style changes severely affecting daily life by COVID-19 countermeasures at their onset?" unclear. Please re-phrase.

7. Please clarify whether the different rows in Table 6 pertain to different outcome variables versus predictor variables. Here I think these are outcomes but I was not sure, please clarify this in all tables. Make it clear in the first column header where the variables below are IVs or DVs.

8. Please re-read carefully for typos. For example, on p.16, line 17; "Thirdly, we run the model" should be "Thirdly, we ran the model". I found a few typos like this scattered throughout the manuscript.

9. P.16, line 46, Hypothesis 3, the term "mediate" was still used inappropriately. Please do a "Command F" search to find any last uses of the term.

Discussion

10. P. 20, line 49, potentially change the phrase "potentially endanger the generalizability" to "potentially compromise the generalizability".

11. In Discussion, on P.21, line 51, I think there is a mistake when the authors say "Overall, respondents perceived actions and countermeasures as more efficacious when they reported higher feelings of control and LOWER perceived risk of contracting the virus" However, the results in Table 4 seem to indicate that perceived risk of the virus predicted GREATER efficacy, not lower.

OSF

12. I found the names of the different data files unclear. It might be helpful to have a clearly labeled final data set for each country (e.g., US_Data) and a clearly labeled data set for the merged file (AllCountries_Data).

Decision letter (RSOS-200644.R1)

Dear Dr Pfuhl:

On behalf of the Editors, I am pleased to inform you that your Manuscript RSOS-200644.R1 entitled "Perceived efficacy of countermeasures and actions and their impact on mental health during the early phase of the COVID-19 outbreak in six countries" has been accepted for publication in Royal Society Open Science subject to minor revision in accordance with the referee suggestions. Please find the referees' comments at the end of this email.

The reviewers and Subject Editor have recommended publication, but also suggest some minor revisions to your manuscript. Therefore, I invite you to respond to the comments and revise your manuscript.

- Ethics statement

- Data accessibility

<http://datadryad.org/submit?journalID=RSOS&manu=RSOS-200644.R1>

- Competing interests

- Authors' contributions

- Acknowledgements

- Funding statement

Because the schedule for publication is very tight, it is a condition of publication that you submit the revised version of your manuscript before 26-Jul-2020. Please note that the revision deadline

will expire at 00.00am on this date. If you do not think you will be able to meet this date please let me know immediately.

on behalf of Dr Christina Demski (Associate Editor)
openscience@royalsociety.org

Associate Editor Comments to Author (Dr Christina Demski):

Associate Editor: 1

Comments to the Author:

Both reviewers have positively reviewed the revised manuscript and suggested it is suitable for acceptance with minor revisions. Both reviewers however still have a list of things that could be improved and as editor I agree that these points should be addressed in full. Please also note that one of the reviewer noted quite a few spelling mistakes etc so please make sure the manuscript is carefully proof-read before submission.

Reviewer comments to Author:

Reviewer: 2

Comments to the Author(s)

This was my second time reviewing the manuscript "Perceived efficacy of countermeasures and actions and their impact on mental health during the early phase of the COVID-19 outbreak in six countries". I was Reviewer 2 in the initial review.

Overall, I think that the revised version of the manuscript is much improved compared to the initial draft, and I appreciate the efforts which the authors took to incorporate my feedback, as well as that of the other reviews. For example, I think that the introduction and discussion are now better organized and more clearly set up the theoretical hypotheses tested in the paper.

This said, there were still a few aspects of the manuscript I was unsure about (some of which are similar to concerns I had in my original review). I describe these below in order of how they appear in the manuscript, but leave it to the editor and the authors to decide what should be implemented or not.

I wish the authors all the best with this important work.

Intro

1. On p.4, lines 36 to 47, I like how the authors note research describing how people are poor predictors of the future and of how they'll actually experience something. However, here the authors seem to assume that people are biased to be overly optimistic about the future, and I wondered if there is any previous research which might suggest that people could also be potentially overly pessimistic?

2. In the introduction, p.5, lines 14-30, the authors make a case for why perceived risk/threat of the virus would predict perceived efficacy of the restrictions used to reduce virus spread. However, here the authors seem to confound acceptability/necessity of the restriction, with the effectiveness of the intervention. But I am not sure if these are one in the same: I might think there is great need to for an intervention/restriction, but also think that the interventions/restrictions presently offered are not effective/efficacious. Perhaps the authors could note this theoretical distinction in the General Discussion, and say how future work could better disentangle these perceptions.

3. Throughout the manuscript I still got confused between the distinction of perceived efficacy of actions (performed by individuals, others, government) and countermeasures proposed by government. This first becomes problematic when understanding how H2 is distinct from H3 on p.7. Moreover, throughout the manuscript, the authors seem to oscillate between them fairly quickly and interchangeably: e.g., see p. 18, line 20: "The majority of our respondents were directly affected by COVID-19 countermeasures. They engaged in protective actions" Here, the authors seem to refer to countermeasures as protective actions, so I am not sure what is what. Maybe, the authors could use more distinct names, or have a paragraph further explaining the distinction between the two sets of constructs?

Study

4. On p. 8, could the authors clarify that for the impact of countermeasures participants rated impact even if they did not experience the countermeasure (i.e., anticipated impact). I think based on results, that this is what is actually done but this is not stated clearly in the manuscript right

now. Also, if participants did not experience something, wouldn't the rating scales have to be framed differently "I would expect this to impact my daily life a lot?".

5. I still am not convinced with how the authors justify using feeling of control operationalized as "I feel we can control the outbreak of the Coronavirus" as a predictor of efficacy of actions operationalized as "My/Our actions are effective in limiting the outgroup". I find that this is the same construct – I would like to know what these items correlate at. Also, assuming they were different, I could see efficacy, thinking actions are effective to limited the diseases spread, predicting (rather than being predicted by) a sense that the country can control the outbreak. Maybe discuss this in the general discussion as a potential limitation.

6. P.12, line 56, I find the phrase: "Were personal life- and work-style changes severely affecting daily life by COVID-19 countermeasures at their onset?" unclear. Please re-phrase.

7. Please clarify whether the different rows in Table 6 pertain to different outcome variables versus predictor variables. Here I think these are outcomes but I was not sure, please clarify this in all tables. Make it clear in the first column header where the variables below are IVs or DVs.

8. Please re-read carefully for typos. For example, on p.16, line 17; "Thirdly, we run the model" should be "Thirdly, we ran the model". I found a few typos like this scattered throughout the manuscript.

9. P.16, line 46, Hypothesis 3, the term "mediate" was still used inappropriately. Please do a "Command F" search to find any last uses of the term.

Discussion

10. P. 20, line 49, potentially change the phrase "potentially endanger the generalizability" to "potentially compromise the generalizability".

11. In Discussion, on P.21, line 51, I think there is a mistake when the authors say "Overall, respondents perceived actions and countermeasures as more efficacious when they reported higher feelings of control and LOWER perceived risk of contracting the virus" However, the results in Table 4 seem to indicate that perceived risk of the virus predicted GREATER efficacy, not lower.

OSF

12. I found the names of the different data files unclear. It might be helpful to have a clearly labeled final data set for each country (e.g., US_Data) and a clearly labeled data set for the merged file (AllCountries_Data).

Reviewer: 1

Comments to the Author(s)

The revised manuscript seems to well-cover its objectives and breadth with the minute modifications carried out. The flow of the writing and presentation is much clear and making a coherent sense. However, few rectifications are still needed before giving it a final touch:

1. The formulation of the hypothesis: Few more deliberations needed for strong positioning of all the hypotheses. The author may include further studies.

2. The Discussion: The discussion segment have been enriched. However, the ground reality behind the differences of perception among men and women in the present sample regarding the efficacy and a more gripping paragraph on findings related to the cultural differences among the two set of countries may yield more findings and extend the depth of the manuscript.

3. On the implication part, the leadership of South Korea and New-Zealand may be cited with the fruitful results as they adhere to trust, transparent and persuasive communication. Role of social media may also be highlighted upon in the case of awareness, minus the fake news.

Author's Response to Decision Letter for (RSOS-200644.R1)

See Appendix B.

Decision letter (RSOS-200644.R2)

Dear Dr Pfuhl,

It is a pleasure to accept your manuscript entitled "Perceived efficacy of COVID-19 restrictions, reactions and their impact on mental health during the early phase of the outbreak in six countries" in its current form for publication in Royal Society Open Science.

You can expect to receive a proof of your article in the near future. Please contact the editorial office (opencience_proofs@royalsociety.org) and the production office (opencience@royalsociety.org) to let us know if you are likely to be away from e-mail contact -- if you are going to be away, please nominate a co-author (if available) to manage the proofing process, and ensure they are copied into your email to the journal.

Kind regards,
Andrew Dunn
Royal Society Open Science Editorial Office
Royal Society Open Science
opencience@royalsociety.org

on behalf of Dr Christina Demski (Associate Editor) and Andrew Dunn (Subject Editor)
opencience@royalsociety.org

Appendix A

Dear Dr. Demski and editorial office,

We thank you and the reviewers for your constructive review and helpful comments. Following your advice, we provide a detailed response to each of the reviewers' comments below. We thank you for the opportunity to submit a revised version of our manuscript, and hope that you find the current version worthy of publication in *Royal Society Open Science*.

Respectfully,
The authors

Reviewer #1

1. However, the weakest part of the paper is its writing style right from the introduction. There are no sub-headings, either in introduction or in the result, leading to more cognitive load on the reader, followed by a weak discussion.

Thank you for pointing this out. We have extensively rewritten the introduction and discussion. We have also added informative subheadings to all sections.

2. The introduction lacks compelling insights-- a brief status report of the COVID-19 in all the six countries should be mentioned one by one with relevant data, so that a reader can have a brief idea of all the six countries.

Thank you for requesting those brief status reports which we previously omitted. We now report the COVID-19 cases and death count for mid-March and the end of March for the six countries, as well as their stringency index.

3. The title of the manuscript state two psychological variables, but, in the variables section, plethora of variables appear without any proper operationalisation, rationalisation, and presentation. It should be properly mentioned as to how and why in the introduction sections, the authors have selected so and so variables. In short, the existing framework fell short in explaining study variables properly and needs refinement. Variables need to be properly defined operationally.

Thank you for raising this point and we agree that we assumed too much from the reader. We now explain the rationale for choosing our measurements, describe the psychological variables in more detail (so that the reader does not need to look them up in the supplementary material) and have restructured the method section with subheadings.

We have also revised our title.

4. Besides, only one question measure feeling of control. What is operationalisation of the construct 'control' in the study?

We thank the reviewer for this excellent question. We ask for “I feel we can control the outbreak of the Coronavirus” on a VAS from 0 (no control) to 100 (full control). We do believe this one question is sufficient. We had to consider the lengths of the survey and preferred to ask directly about feeling of control rather than using a scale. We have also clarified our construct of control by referring to it as controlling the outbreak (see Methods section).

5. The result section is appear vague and with certain wrong interpretations. For ex - the last line of Page 9, it is mentioned that feeling of control and perceived risk was similar Germany and US, but the data in the table reflects differently. Similarly, on page 10, worry and perceived risk was higher than perceived risk, except Norway, which is not in sync with the data. In spite of the data being rich, putting them in a single-go, make it hard for the reader to comprehend the meaning easily. The way hypothesis have been derived is not properly written.

We imprecisely formulated the description of some of the data. We have now corrected these issues throughout the results section.

6. The discussion is too generic and lacks compelling arguments, reasoning, and theoretical underpinnings. For ex - Even if the gender and age has been considered, the reason for their findings were not provided. Given the study encompasses six countries, country-specific cultures can also play a huge role for the findings and need explicit mentioning in the discussion.

We have extensively edited the discussion section. We now provide theoretical underpinnings for our findings. We further speculate about the observed cross-country differences, given that we did not recruit a representative sample from each country.

Reviewer #2

7. It was not clear to me why some of the variables were being included either as predictors or DVs. For example, in Table 3 and Table 4 the authors regress several factors onto perceived efficacy of actions. I believe that in both Table 3 and 4 this was the combined efficacy score of personal actions, country actions, and national actions although I was not sure because in Table 3 title is just said “efficacy of actions” while in Table 4 it explicitly said personal/country/other. I can understand why personal control might relate to perceived efficacy of individual actions – although the authors need to review more literature on why this would be true – but I am less clear on why personal control would predict perceived efficacy of the nation/or others’ actions. Even if I have a

high sense of personal control, I might think that my country is not effective? Other variables which were included as predictors were also less clear to me. For example, why would my level of paranoia relate to my perceived efficacy of actions like social distancing? For every predictor that is included in the regression models, it will help for the authors to justify on the basis of past literature why it would / or would not be expected to relate to the DV being predicted.

We have extensively edited the introduction to provide clear theoretical justifications for each of the variables we have included as predictors. Furthermore, we now clearly indicate analyses in which we refer to the combined score of efficacy of personal/other/governmental actions or to each of them separately.

8. I had similar questions about the justification of predictors included to predict psychological distress. For one, I wondered if worry/fear is itself an index of distress? Can the authors provide justification for why these two variables should be treated separately? Similarly, the clinical measure of paranoia seemed somewhat related to distress as well, or at least, should be treated more as a DV? If we look at the effect sizes of these two predictors of general distress they are also markedly higher than the other predictors, which added to my concern about their distinctions from the DV. Moreover, in the introduction, p.3, lines -31-34 the authors note briefly “anxiety and paranoia have been shown to develop under conditions of worry and perceived loss of control”. Here, paranoia seems to be presented as a DV developing during conditions of uncertainty. Also, it seems a little circular to me to state that anxiety develops from worry because worry is a core component of anxiety? Paranoia was also described as a DV on p. 56 where the authors write: “As the COVID-19 outbreak poses a severe threat which can cause physical, psychological and social harm, it can enhance long-term anxiety, which, in turn, can lead to maladaptive outcomes such as paranoia”.

We found that worry/fear were directly linked to the pandemic, whereas general distress was not. We believe that general distress can be driven by perceived risk of contagion and or worry/fear of own contagion. A person might judge the risk of infecting others high but not worried about their own health and vice versa, both extreme cases might lead to elevated feelings of distress. Further, one can be worried about terrorism but this may have no effect on one’s general distress. Worry/fear about COVID-19 is therefore not the same as general distress. We have clarified this now in the manuscript.

We agree that paranoia can be treated as a DV too, however, we did not pre-register this and therefore refrain from reporting too many exploratory analyses. We have rewritten the section on p. 5

9. I also wondered how perceived knowledge about the pandemic and belief in conspiracy theories fit into the authors overarching theoretical framework. Can they provide justification for why it makes sense to compute a difference score of knowledge about COVID-19 from perceived conspiracies? Does past literature do this? I imagine that I might know very little about a subject, but also, be very skeptical of conspiracy theories. I would want to also see a review/discussion of past literature describing the link

between knowledge/conspiracy theories and perceived efficacy of self/other/government actions and distress. Why should we expect knowledge/conspiracy theories to relate to these outcomes? I am guessing that the authors conceptualize this measure as an index of uncertainty (a construct they discuss on p.3 lines 31-36) but this could be made clearer.

We appreciate these questions. We have edited the introduction to include the literature showing that people who believe in conspiracy theories do perceive actions of their government as less efficient (for preprints on COVID-19 about this topic please see: Pennycook, G., McPhetres, J., Bago, B., & Rand, D. (2020). Predictors of attitudes and misperceptions about COVID-19 in Canada, the UK, and the USA; Alper, S., Bayrak, F., & Yilmaz, O. (2020). Psychological Correlates of COVID-19 Conspiracy Beliefs and Preventive Measures: Evidence from Turkey. <https://psyarxiv.com/mt3p4>).

Regarding calculating a difference score: We calculated a difference score because we are interested in measuring a participant's ability to distinguish between factual and non-factual statements about COVID-19. This is common practice for e.g. distinguish bullshit, confabulations (Pennycook et al., 2015, Mækele, Moritz, Pfuhl, 2018), and if one uses more items one can calculate a discrimination score based on Signal Detection theory.

Our main interest was in how much a person was correctly informed about the Coronavirus. Ignorance may yield a neutral score (neither scoring high on knowledge items nor on conspiracy items), favoring conspiracy items rarely coincides with endorsing factual knowledge (e.g. Pennycook, G., Fugelsang, J. A., & Koehler, D. J. (2015). Everyday consequences of analytic thinking. *Current Directions in Psychological Science*, 24(6), 425-432; Swami, V., Voracek, M., Stieger, S., Tran, U. S., & Furnham, A. (2014). Analytic thinking reduces belief in conspiracy theories. *Cognition*, 133(3), 572-585; Pennycook, G., McPhetres, J., Zhang, Y., & Rand, D. (2020). Fighting COVID-19 misinformation on social media: Experimental evidence for a scalable accuracy nudge intervention. *PsyArXiv Preprints*, 10). We therefore kept the knowledge score.

10. I think a lot of the concerns I raised above can be addressed by re-working the introduction and the theoretical framework presented there. First, I think there is some literature reviewed here which does not really relate to what was measured in the actual research. Most notably is the discussion on p.3 to p.4 about uncertainty leading to pro-sociality - I was confused about this, because I don't think the authors measure pro-sociality.

We omitted the paragraph describing potential beneficial consequences related to abrupt changes in everyday life, and replaced it with the following sentence: "for potential positive consequences such as increased prosociality, see. e.g., Aknin, Dunn, Whillans, Grant, & Norton, 2013; Kappes et al., 2018".

11. In the introduction, it would be helpful for the authors to clearly operationalize and define all the IVs and DVs they will focus on, and explain why they would be expected to relate. For example, the authors talk a lot about “control” but they never really define it. Psychological control has a broad literature associated with it, and has been defined in different ways (e.g., thinking I can influence or control others, or resist influence from others; internal/external locus of control ect). Perceptions of control are also sometimes operationalized as part of efficacy/competence. So, it will be helpful to know exactly what the authors mean by control, and how in their framework it is distinct from efficacy. Also, the authors frequently discussed uncertainty stemming from a pandemic, but they never explicitly say they measured uncertainty in their research. Thus, I want to know how uncertainty might have been operationalized in their work (if it was): perhaps it was the knowledge measure? , or the paranoia measure?, or the worry/fear measure? All of this needs to be made more clear.

Thank you for requesting those clarifications. We now describe control as “feeling we can control the outbreak” which is the original formulation of the item. Thus, control refers to a perception of efficacy without specifying how the control happens (countermeasures) or who is doing it (own, other, government).

Furthermore, we have rewritten the introduction to avoid the theoretically broad term of uncertainty.

12. Lastly, I think it could be critical for the authors to flesh out the four predictions they quickly summarize on p.4 (lines 54-59) and then re-state as hypotheses on p.6-7. First, I suggest that these two sections could be merged into one section and just appear at the end of the introduction. Second, there is never any justification of “why” provided for any of these predictions. For example, why would we expect that people who are actually experiencing countermeasures will perceive them as more impactful, then how people who have not yet experienced them might imagine them to be?

We have combined the sections according to the helpful suggestion by the reviewer (see page 7). We have also significantly modified the introduction to provide broader context for our predictions.

13. The authors often use the term “mediate” but I am not sure this term is being used correctly in the paper. They seem to be using the term to mean “predict”. For example, on p. 13 they write “ which factors mediate the participants’ perceived efficacy of the countermeasures?”. But mediation refers to one variable “M” explaining/underlying the relation between to other variables X and Y. Its not clear to me what X,M,Y are in this case.

We have rewritten references to mediation throughout the manuscript.

14. I was confused by the authors' decision to measure both perceived efficacy of government actions and also countermeasures initiated by the government. Are these not the same thing? The authors themselves even say something to this extent on p. 7, line 58 " We did not include the efficacy of countermeasures in this analysis to avoid redundancy, as the perceived efficacy of actions score includes actions by government". Why measure two very similar things, and use one versus the other across different analyses?

We now better differentiate the two variables (see Variables and Indices section). In the section descriptives we inserted a correlation plot (previously in the supplementary material) showing the difference between the specific countermeasures and action agents (self, other, government). Furthermore, perceived efficacy of (governmental) actions measures the general impression participants have formed on the efficacy of actions, whereas efficacy of countermeasures pertains to the perceptions of specific countermeasures enacted by the government. We have also clarified the reason for not including the efficacy of countermeasures score in the aforementioned analysis (p 10): "We did not include the efficacy of countermeasures in this analysis, as the perceived efficacy of actions score, although general in its nature, can be impacted by the impression of specific actions by the government."

15. As I noted above I think the decision to combine personal, other, government efficacy into one measure to test hypothesis 2 needs more justification. I could see these being quite distinct, and indeed, these are what the descriptive results presented by the authors shows.

We now report the results of the post-hoc analyses of perceived efficacy of actions separately for personal, other and governmental actions. As the results indicate, the different predictors had generally comparable contribution to all three kinds of actions.

16. I would like to see more consistency in how the variables are referred to across the different tables. For example, sometimes the authors wrote CORE-9/distress (Table 2) and other times just CORE-9 (Table 3. Also, why is CAPE-P not referred to as CAPE-P/paranoia in Table 2, similar to CORE-9/distress? Personally, I think it is easier to follow when variables are referred to by the construct name (e.g., distress, paranoia) versus the scale name (CORE-9, CAPE-P).

We have edited the manuscript according to the reviewer's suggestion. We now refer to variables by the construct name rather than the scale name.

17. In the results section, I appreciated how the authors reminded us what each hypothesis was : e.g., " Hypothesis 4: Which factors contribute to general distress among participants?" but I think it may be easier to follow if they frame this as an actual hypothesis, (A,B,C will predict general distress) versus a question.

We now provide clear subheadings (repeating the hypothesis) and have therefore not framed it as a question.

18. On p. 9, I was confused by this statement: “On the other hand, 235 (10.4%) indicated that they did not perform any protective actions, and 201 (9.2%) answered that they did not perform any action to protect others. There were 111 participants doing neither.” How could there be 235 people who did not perform any protective actions, but then also, 111 participants doing neither?

We have clarified the text to better indicate that 235 participants were not taking any actions to protect themselves, and 111 participants were not taking any actions to protect themselves and others. The text now reads: “On the other hand, 235 (10.4%) indicated that they did not perform any protective actions to protect themselves from the virus, and 201 (9.2%) answered that they did not perform any action to protect others. Of these participants, a total of 111 participants were doing neither.”

19. I found the variable name “effect of countermeasures” confusing – this sounds a lot like “impact” of countermeasures. Maybe relabel to “Types of countermeasures initiated by government”. It would also be helpful to explicitly say that participants could select more than one option here.

We have relabeled the variable “experienced countermeasures”, and we now mention that participants could choose multiple items in response to this question.

20. I am not sure that the Tables are using APA style

RSOS does not specify the table style but we have now aligned them with APA style.

Appendix B

Reply to decision letter / reviewers

We appreciated highly the very constructive feedback from reviewer 2. All points raised have been fully addressed by us. Details described below in italic.

Reviewer: 2

Intro

1. On p.4, lines 36 to 47, I like how the authors note research describing how people are poor predictors of the future and of how they'll actually experience something. However, here the authors seem to assume that people are biased to be overly optimistic about the future, and I wondered if there is any previous research which might suggest that people could also be potentially overly pessimistic?

Reply: We now also cite a paper finding both optimism and pessimism

2. In the introduction, p.5, lines 14-30, the authors make a case for why perceived risk/threat of the virus would predict perceived efficacy of the restrictions used to reduce virus spread. However, here the authors seem to confound acceptability/necessity of the restriction, with the effectiveness of the intervention. But I am not sure if these are one in the same: I might think there is great need to for an intervention/restriction, but also think that the interventions/restrictions presently offered are not effective/efficacious. Perhaps the authors could note this theoretical distinction in the General Discussion, and say how future work could better disentangle these perceptions.

Reply: Thank you for drawing our attention to this highly valuable distinction. We do discuss it, please see page 19, new paragraph on acceptance vs perceived efficacy of restrictions and reactions

3. Throughout the manuscript I still got confused between the distinction of perceived efficacy of actions (performed by individuals, others, government) and countermeasures proposed by government. This first becomes problematic when understanding how H2 is distinct from H3 on p.7. Moreover, throughout the manuscript, the authors seem to oscillate between them fairly quickly and interchangeably: e.g., see p. 18, line 20: "The majority of our respondents were directly affected by COVID-19 countermeasures. They engaged in protective actions" Here, the authors seem to refer to countermeasures as protective actions, so I am not sure what is what. Maybe, the authors could use more distinct names, or have a paragraph further explaining the distinction between the two sets of constructs?

Reply: We now use the terms restrictions (previously countermeasures) and reactions (previously actions). Note that protective actions like handwashing are still referred to as actions.

Study

4. On p. 8, could the authors clarify that for the impact of countermeasures participants rated impact even if they did not experience the countermeasure (i.e., anticipated impact). I think based on results, that this is what is actually done but this is not stated clearly in the manuscript right now. Also, if participants did not experience something, wouldn't the rating scales have to be framed differently "I would expect this to impact my daily life a lot?".

Reply: We have clarified this. Thanks

5. I still am not convinced with how the authors justify using feeling of control operationalized as “I feel we can control the outbreak of the Coronavirus” as a predictor of efficacy of actions operationalized as “My/Our actions are effective in limiting the outbreak”. I find that this is the same construct – I would like to know what these items correlate at. Also, assuming they were different, I could see efficacy, thinking actions are effective to limited the diseases spread, predicting (rather than being predicted by) a sense that the country can control the outbreak. Maybe discuss this in the general discussion as a potential limitation.

Reply: We clarify it by referring to self-efficacy for own reactions and “we can control” as more general efficacy. We report the correlation between the two items and do also discuss this.

6. P.12, line 56, I find the phrase: “Were personal life- and work-style changes severely affecting daily life by COVID-19 countermeasures at their onset?” unclear. Please re-phrase.

Reply: Rephrased to “Direct impact of restrictions on daily life”

7. Please clarify whether the different rows in Table 6 pertain to different outcome variables versus predictor variables. Here I think these are outcomes but I was not sure, please clarify this in all tables. Make it clear in the first column header where the variables below are IVs or DVs.

Reply: We clarified this. Since it is a logistic regression, the rows are predictors = IVs

8. Please re-read carefully for typos. For example, on p.16, line 17; “Thirdly, we run the model” should be “Thirdly, we ran the model”. I found a few typos like this scattered throughout the manuscript.

Reply: We have carefully read the manuscript for past tense and corrected it where appropriate.

9. P.16, line 46, Hypothesis 3, the term “mediate” was still used inappropriately. Please do a “Command F” search to find any last uses of the term.

Reply: Thanks, it was the only instance.

Discussion

10. P. 20, line 49, potentially change the phrase “potentially endanger the generalizability” to “potentially compromise the generalizability”.

Reply: Agree, we replaced endanger with compromise

11. In Discussion, on P.21, line 51, I think there is a mistake when the authors say “Overall, respondents perceived actions and countermeasures as more efficacious when they reported higher feelings of control and LOWER perceived risk of contracting the virus” However, the results in Table 4 seem to indicate that perceived risk of the virus predicted GREATER efficacy, not lower.

Reply: We corrected that mistake.

OSF

12. I found the names of the different data files unclear. It might be helpful to have a clearly labeled final data set for each country (e.g., US_Data) and a clearly labeled data set for the merged file (AllCountries_Data).

Reply: We have now created a datafile AllCountries_March2020 and uploaded it into the Analysis folder. The raw data files are by language not by country and contain the name of the survey in the title, the language and the date of retrieving it from Qualtrics. We think this is the most transparent way. Due to a change in measuring the item "country" we uploaded a raw data file in xls format containing raw data for all but the country item to avoid confusion. This change is noted in the osf page wiki too.

Reviewer: 1

Comments to the Author(s)

The revised manuscript seems to well-cover its objectives and breadth with the minute modifications carried out. The flow of the writing and presentation is much clear and making a coherent sense. However, few rectifications are still needed before giving it a final touch:

1. The formulation of the hypothesis: Few more deliberations needed for strong positioning of all the hypotheses. The author may include further studies.

Reply: We have substantiated each of our hypothesis with 2-4 references. Some of the articles we refer to have been published after we pre-registered our analysis

2. The Discussion: The discussion segment have been enriched. However, the ground reality behind the differences of perception among men and women in the present sample regarding the efficacy and a more gripping paragraph on findings related to the cultural differences among the two set of countries may yield more findings and extend the depth of the manuscript.

Reply: We briefly discuss gender disparity, and extended possible cultural differences, substantiated by very recent publications.

3. On the implication part, the leadership of South Korea and New-Zealand may be cited with the fruitful results as they adhere to trust, transparent and persuasive communication. Role of social media may also be highlighted upon in the case of awareness, minus the fake news.

Reply: Thank you, we expanded on trust and leadership, and cite relevant (recent) references.